# CoT-Space: A Theoretical Framework for Internal Slow-Thinking via Reinforcement Learning

## Abstract

Reinforcement Learning (RL) has become a pivotal approach for enhancing the reasoning capabilities of Large Language Models (LLMs). However, a significant theoretical gap persists, as traditional token-level RL frameworks fail to align with the reasoning-level nature of complex, multi-step thought processes like Chain-of-Thought (CoT). To address this challenge, we introduce CoT-Space, a novel theoretical framework that recasts LLM reasoning from a discrete token-prediction task to an optimization process within a continuous, reasoning-level semantic space. This shift in perspective serves as a conceptual bridge, revitalizing foundational principles from classical learning theory to analyze the unique dynamics of LLMs. By analyzing this process from both a noise perspective and a risk perspective, we demonstrate that the convergence to an optimal CoT length is a natural consequence of the fundamental trade-off between underfitting and overfitting. Furthermore, extensive experiments provide strong empirical validation for our theoretical findings. Our framework not only provides a coherent explanation for empirical phenomena such as overthinking but also offers a solid theoretical foundation to guide the future development of more effective and generalizable reasoning agents. We open-source our code through an anonymous GitHub repository at `https://anonymous.4open.science/r/CoT-Space-Reasoning-via-RL`.

## 1 Introduction

Improving the reasoning capabilities of Large Language Models (LLMs) is pivotal in modern AI research. The most well-known framework for this is Chain-of-Thought (CoT) (Wei et al., 2022b), which elicits multi-step reasoning. A promising strategy to enhance CoT is "slow-thinking" (Chen et al., 2025b; Jiang et al., 2024), which can be broadly categorized into external and internal methods. External methods, such as tree-based strategies (Yao et al., 2023; Zhang et al., 2024; Wan et al., 2024) and self-consistency (Wang et al., 2022), augment the model's inference-time output via searching (Gan et al., 2025). Unlike external methods that rely on explicit search algorithms, internal slow-thinking embeds the reasoning process directly into the model's forward pass via post-training, typically utilizing Reinforcement Learning (RL) with a high-level outcome reward.

Recent breakthroughs from models like DeepSeek's R1 (DeepSeek-AI, 2025), OpenAI's o1 and o3 (OpenAI, 2024; 2025), and Qwen's QwQ (Qwen, 2024) have spotlighted the power of using RL for internal slow thinking. This paradigm, often called *zero*-like training, involves applying RL directly to a pre-trained model, bypassing SFT. The results are compelling, showing that LLMs can learn sophisticated reasoning strategies through self-exploration. This aligns with a core tenet of RL: a high-level objective can guide a policy to discover an optimal strategy on its own.

As illustrated in Figure 1, RL for LLM reasoning is analogous to classical RL scenarios. In a classic RL task like Bricks Breaker[1] shown in subgraph (a), the agent's goal is to break more bricks. Without explicit instructions on specific strategies, the agent may discover a highly effective method, such as hitting the ball behind the top row of bricks to achieve a higher score. This kind of emergent behavior is a common outcome in classical RL. Similarly, in the context of LLM reasoning, as shown

---

[1]Bricks Breaker is a video game, the goal is to destroy the bricks by shooting a ball at them.

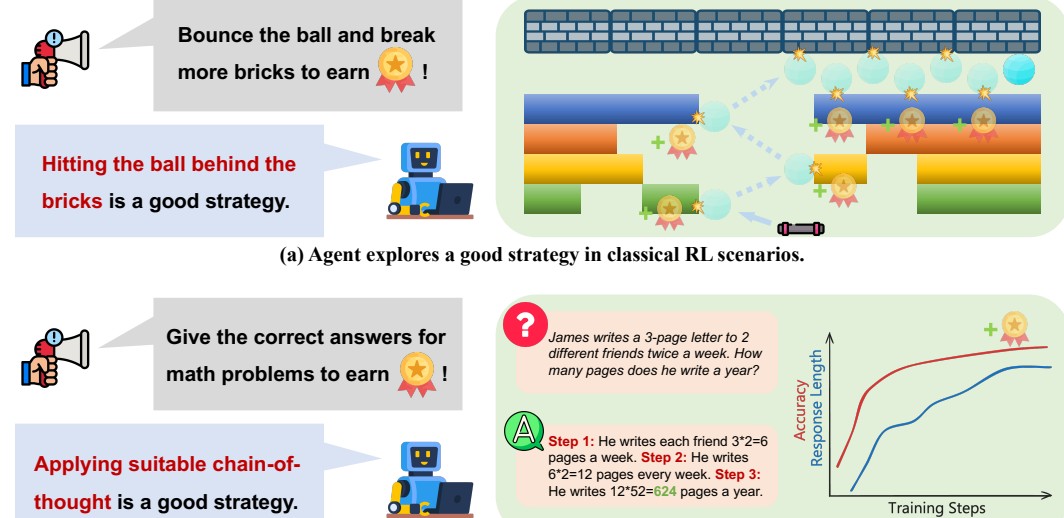

(a) Agent explores a good strategy in classical RL scenarios.

(b) Policy explores a good strategy in RL for LLM reasoning scenarios.

Figure 1: **Analogy between strategy discovery in classical RL and LLM reasoning.** (a) In classical RL, an agent with a high-level goal (e.g., break more bricks) discovers an effective strategy through exploration to maximize its reward. (b) Similarly, an LLM policy with the goal of providing correct answers autonomously learns that generating a suitable CoT is an effective strategy.

in subgraph (b), the policy model is given the broad goal of providing a correct answer. As RL training progresses, the model converges on an effective CoT pattern, demonstrating its ability to learn complex reasoning pathways from a simple, high-level reward signal.

Building on this, a variety of RL-based methods have been developed to improve *zero*-like post-training, including GRPO (Shao et al., 2024), VC-PPO (Yuan et al., 2025), DAPO (Yu et al., 2025), and VAPO (Yue et al., 2025b). However, despite these empirical successes, our theoretical understanding of internal slow thinking remains shallow. For instance, the "overthinking" phenomenon (Sui et al., 2025), which suggests an optimal CoT length exists, lacks a clear mechanistic explanation. Furthermore, the principles governing the generalization of LLM reasoning are not yet well understood.

This chasm between empirical success and theoretical understanding is alarming. It stands in stark contrast to classical machine learning, where decades of research have forged a robust theoretical bedrock. Foundational principles like optimization theory and generalization bounds, the very cornerstones of traditional ML, appear to falter when applied to the complex, multi-step nature of LLM reasoning. This prompts the critical question that motivates our work: **Are these time-tested theories truly obsolete in this new era, or do we merely lack the conceptual framework to bridge them to the unique dynamics of LLM reasoning?** We argue for the latter, positing that by shifting our analytical perspective, we can revitalize these foundational theories and use them to build a principled understanding of LLM reasoning.

The central barrier to this goal is twofold. **First, a fundamental misalignment exists between token-level RL frameworks and the reasoning-level nature of CoT**, where actions are complete thoughts rather than single tokens. **Second, a conceptual gap separates the discrete world of language from the continuous mathematics underpinning classical learning theory**, hindering the application of its powerful analytical tools. To resolve both challenges, we introduce **CoT-Space**, a novel reasoning-level theoretical framework for LLM reasoning via RL. We emphasize that CoT-Space is primarily a conceptual framework. Our goal is not to propose a computationally executable algorithm for extracting reasoning steps, but rather to build a theoretical bridge that connects the empirical phenomena of LLM reasoning with foundational principles of classical machine learning. Our framework first defines a reasoning-level state space to align RL with CoT's structure. Subsequently, we prove this space converges to a continuous manifold, a key result that recasts reasoning as an optimization process. By leveraging this continuous perspective, we are then able to conduct analyses that explain the convergence of CoT length and the generalization of reasoning, revitalizing classical theory to address the key theoretical gaps previously identified.

The remainder of this paper is structured as follows. In Section 2, we provide a systematic analysis of the misalignment between the current token-level RL formulation and the reasoning-level CoT paradigm. Subsequently, we introduce our reasoning-level theoretical framework, CoT-Space. We then leverage this framework in Section 3 to analyze the convergence of CoT length in the reasoning process from the perspective of noise and risk, demonstrating its potential theoretical value. We perform empirical validation of our theoretical insights in Section 4. Following this, we briefly review related works in Section 5 and finally conclude the paper in Section 6.

## 2 CoT-Space: A Reasoning-Level Theoretical Framework

This section presents a systematic analysis of CoT reasoning via RL. We first discuss this misalignment (Subsection 2.1), then introduce our reasoning-level framework, CoT-Space (Subsection 2.2), and finally prove its continuum nature to establish its theoretical foundation (Subsection 2.3).

### 2.1 From Token-Level to Reasoning-Level Perspective

The conventional theoretical framework for RL in LLMs is fundamentally misaligned with the reasoning-level nature of CoT scenarios. This misalignment stems from modeling auto-regressive language generation as a token-level Markov Decision Process (MDP) (Setlur et al., 2025): $\mathcal{M}(\mathcal{S}, \mathcal{A}, r, H)$, where $\mathcal{S}$ is the state space of existing token sequences, $\mathcal{A}$ is the action space representing the token dictionary, $r$ is the reward function guiding generation, and $H$ is the token budget. In this standard formulation, a policy $\pi : \mathcal{S} \to \mathcal{A}$ produces the next token $a = \pi(s)$ to extend the current sequence $s$. While inherited from classic RL, this approach presents a significant challenge regarding the token-level state space $\mathcal{S}$ in natural language, as we assert next.

**Assertion 2.1.** *The token-level state space $\mathcal{S}$ of natural language exhibits both **exponential growth** and **discrete topology**.*

To address this, we propose shifting the analytical perspective from the microscopic token level to the macroscopic reasoning level. We posit that while the token-level state space is discrete and sparse, the reasoning-level state space, which is formed by abstract thoughts, can be approximated as a continuous manifold. This is analogous to interstellar travel: while individual stars (tokens) are discrete, the galaxy (semantic space) appears continuous from a distance. This "zoom-out" perspective allows us to apply continuous optimization tools to analyze the reasoning process, as discussed in Figure 2.

### 2.2 Formulation of CoT-Space

To formalize this conceptual "zoom out" framework, we define the following constructs. Note that these definitions rely on abstract semantic equivalence, serving as theoretical tools rather than computable functions.

Based on the CoT setting, where the output is generated step-by-step and consists of multiple intermediate thoughts, a reasoning state is defined as follows.

**Definition 2.2.** *(CoT reasoning state.) Given a query $q$ and an incomplete set of intermediate reasoning steps $\boldsymbol{\xi}_o = [\xi_1, \xi_2, \ldots, \xi_t]$, the current reasoning state is defined as $s_o = (q, \boldsymbol{\xi}_o)$.*

Definition 2.2 establishes the concept of reasoning-level state. We then introduce a special class of states that represent successful problem solutions, which we call *minimums*.

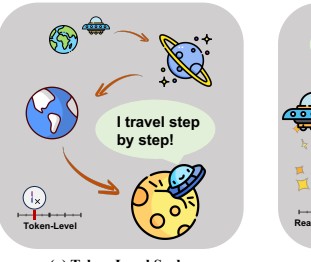
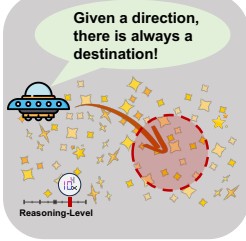

(a) Token-Level Scale    (b) Reasoning-Level Scale

Figure 2: **Token-level vs. Reasoning-level perspectives.** (a) The token-level view treats the generation process as a path through discrete states. (b) The reasoning-level view zooms out, approximating the dense state space as continuous.

**Definition 2.3.** *(Minimums in reasoning space.) Given a query $q$ with a definite golden answer $\phi$. Assume $\Xi_q$ is the set of all reasonable intermediate reasoning processes for query $q$. Then the minimums of the reasoning space are defined as $\boldsymbol{M}_q = \{(q, \boldsymbol{\xi}, \phi) \mid \boldsymbol{\xi} \in \Xi_q\}$.*

According to Definition 2.3, minimums are special states where the query has been successfully solved. For any intermediate reasoning state, we can then define which minimums are reachable.

**Definition 2.4.** *(Reachable minimums.) Given a query $q$ with a definite golden answer $\phi$, and an incomplete set of intermediate reasoning steps $\boldsymbol{\xi}_o = [\xi_1, \xi_2, \dots, \xi_t]$. The reachable minimums are defined as $\boldsymbol{R}_q^o = \{(q, \boldsymbol{\xi}, \phi) \mid \boldsymbol{\xi}_o \sqsubseteq \boldsymbol{\xi} \text{ and } \boldsymbol{\xi} \in \Xi_q\}$, where $a \sqsubseteq b$ represents that $a$ is prefix of $b$.*

Definition 2.4 established the concept of reachable minimums. This concept represents the set of successful reasoning paths that can be completed from the current state. Building on this, we quantify the distance between a state and its reachable minimums and define the nearest minimum.

**Definition 2.5.** *(Reachable distance and nearest reachable minimum.) Given a query $q$ with a definite golden answer $\phi$, and a reasoning state $s_o$ on this query, the reachable distance between the $s_o$ and a reachable minimum is defined as $\text{dist}(s_o, m) = |\boldsymbol{\xi}| - |\boldsymbol{\xi}_o|$, where $m \in \boldsymbol{R}_q^o$, $\boldsymbol{\xi}$ and $\boldsymbol{\xi}_o$ are intermediate steps of $m$ and $s_o$ respectively, $|\cdot|$ represents the length of a vector. The nearest minimum is defined as $m_o^* = \arg\min_{m \in \boldsymbol{R}_q^o} \text{dist}(s_o, m)$.*

Definition 2.5 quantifies the number of reasoning steps required to reach the closest successful solution from a given state. Finally, we define a reasoning loss to characterize the gap between the current state and the optimal solution.

**Definition 2.6.** *(Reasoning loss.) Given a query $q$ with a definite golden answer $\phi$, and a reasoning state $s_o$ on this query, there exists a loss function $C$ that quantifies the distance of $s_o$ towards its nearest reachable minimum, which satisfies: for two different states $s_i, s_j$, $C(s_i) < C(s_j)$ if and only if $\text{dist}(s_i, m_i^*) < \text{dist}(s_j, m_j^*)$. Specially, $\forall m \in \boldsymbol{M}_q, C(m) = 0$.*

Definition 2.6 conceptualizes the reasoning process as an optimization process. The function $C(\cdot)$ acts as a proxy for the distance to a correct solution, where a lower loss indicates a more advanced stage of reasoning. Taking Figure 3(a) as instance, in this transition, $m_i, m_j$ and $m_k$ are all reachable minimums for the current state $s_0$. However, the possible next states of $s_0$, which are denoted as $s_i, s_j$ and $s_k$ (Here we refer to the three realizations $s_{i_1}, s_{i_2}$ and $s_{i_3}$ all as $s_i$ for simplicity). These intermediate states differ in their distance to its corresponding minimums ($\text{dist}(s_i, m_i) < \text{dist}(s_j, m_j) < \text{dist}(s_k, m_k)$), thus the reasoning losses satisfy $C(s_i) < C(s_j) < C(s_k)$.

Before proceeding, it is crucial to clarify the relationship between our theoretical loss $C(s)$ and the practical metrics used in our experiments. In our framework, $C(s)$ represents the "distance to solution," where $C(m) = 0$ for a correct answer. In our RL experiments, this corresponds to maximizing a task-level reward (e.g., $+1$ for success). Consequently, minimizing the expected reasoning loss is mathematically equivalent to maximizing the success rate (Accuracy), which serves as our primary evaluation metric. Although the theoretical loss landscape is continuous, the RL agent navigates it by learning a value function that approximates the expected future reward, effectively performing a stochastic descent on this unobserved loss surface to improve accuracy.

With this complete CoT-Space formulation, we resolve the misalignment between token-level RL and the reasoning-level CoT paradigm. Consider the math problem in Figure 1 ($q$: "James writes..."), We here present an example for better clarification. State ($s_t$): The model has generated "Step 1: He writes 6 pages a week."Step ($\xi$): The abstract logical move of "calculating weekly pages." Token Realization ($K$): The specific tokens "3 * 2 = 6". Minimum ($m \in M_q$): A future state where the final answer "624 pages/year" is reached. Reasoning Loss ($C(s)$): A theoretical measure of how many logical steps remain to reach the answer. We now turn to prove the continuum nature of reasoning-level state space, thus yielding the rationality of CoT-Space.

## 2.3 ON THE CONTINUUM NATURE OF THE REASONING-LEVEL STATE SPACE

The essence of reasoning lies in the semantic meaning of conceptual steps, not in the specific tokens used to express them. A single reasoning step, such as "calculating the total weekly pages" can be realized by a vast number of distinct token sequences $\xi_{i_1}, \xi_{i_2}$ and $\xi_{i_3}$, as shown in Figure 3(a).

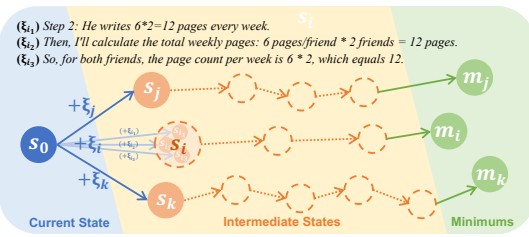 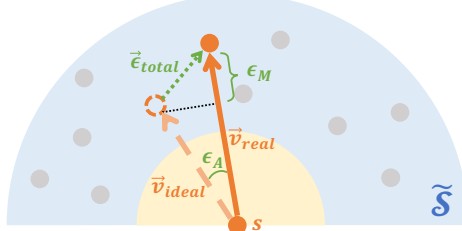

**(a) Reasoning-Level State Transition**  **(b) Continuum Approximation**

Figure 3: **Illustration of CoT-Space.** (a) Different reasoning steps $(\xi_i, \xi_j, \xi_k)$ lead to different minimums $(m_i, m_j, m_k)$, while one step can have multiple token-level realizations $(\xi_{i_1}, \xi_{i_2}, \xi_{i_3})$. (b) The discrete state space $\mathcal{S}$ (dots) is approximated as a continuous manifold $\tilde{\mathcal{S}}$ (blue area), where the real optimization vector $(\vec{v}_{\text{real}})$ approximates the ideal one $(\vec{v}_{\text{ideal}})$.

From token-level perspective, different token sequences correspond to distant, discrete states. However, in a higher-dimensional semantic space, they are nearly identical. The CoT-Space framework posits that as the reasoning length increases, the number of semantically meaningful states grows so rapidly that the space they inhabit can be treated as a continuous manifold. This density arises from the expressive redundancy of language: the number of valid token-level paths to achieve a reasoning goal grows exponentially with the available token budget.

To formalize this, we introduce the following definitions and a key assumption.

**Definition 2.7.** *(Semantic equivalence set.) For an abstract reasoning step $\xi_l$ (e.g., a calculation, a logical deduction), its semantic equivalence set $\mathcal{V}(\xi_l, k)$ is the collection of all token sequences $\tau \in \mathcal{A}^k$ with length $k$ that are semantically equivalent to $\xi_l$, defined as: $\mathcal{V}(\xi_l, k) = \{\tau \in \mathcal{A}^k \mid Decode(\tau) \equiv \xi_l\}$, where $\mathcal{A}^k$ is the space of all possible token sequences with length $k$, and $Decode(\cdot)$ is an abstract function mapping a token sequence to its semantic meaning.*

**Definition 2.8.** *(Reasoning state density.) Within a fixed semantic volume $\mathcal{V}_{semantic}$, the set of all valid, reachable reasoning states of reasoning token amount $K$ is denoted by $\mathcal{S}_{reasoning}^{(K)}$. The state density $\rho(K)$ is defined as the number of states per unit of semantic volume: $\rho(K) = |\mathcal{S}_{reasoning}^{(K)}| / |\mathcal{V}_{semantic}|$.*

To substantiate our claim that this density grows exponentially, we propose the following assumption.

**Assumption 2.9.** *(Exponential expressive redundancy of language.) For any non-trivial reasoning step $\xi_l$, the size of its semantic equivalence set $|\mathcal{V}(\xi_l, k)|$, when realized by $k$ tokens, grows exponentially with $k$. That is, there exists a constant $c > 1$ such that: $|\mathcal{V}(\xi_l, k)| \geq c^k$.*

This assumption acts as a foundational theoretical posit for our framework, grounded in the combinatorial nature of language where adding tokens exponentially increases the space of semantically equivalent expressions. While treated here as a structural condition, we acknowledge that this hypothesis invites empirical validation, for instance, by measuring nearest-neighbor distances of paraphrased reasoning steps in a semantic embedding space. We discuss this potential validation methodology in detail in Appendix E.4.

Based on these foundations, we can now formally state the theorem that justifies the continuum approximation of reasoning-level state space.

**Theorem 2.10.** *(Continuum convergence of reasoning-level state space.) Under Assumption 2.9, in a $D$-dimensional semantic space, the reasoning-level state density, denoted as $\rho(K)$, increases exponentially with the reasoning token amount $K$, formally $\rho(K) = \Theta(c^K)$. Let $s_i \bowtie s_j$ denotes that $s_i, s_j$ are nearest neighbor of each other, the expected nearest neighbor distance of the semantic manifold is inversely related to the reasoning state density, formally $\mathbb{E}_{s_i \bowtie s_j}[\text{dist}(s_i, s_j)] \propto \rho(K)^{-1/D}$.*

The proof is provided in Appendix B.1. In Theorem 2.10, we established that the reasoning-level state space converges to a continuous semantic manifold in a macroscopic limit. Here, we further quantify the total "gap" between an ideal continuous update and the best possible discrete step, demonstrating that this gap vanishes as the total reasoning token amount $K$ increases.

A visualization for this analysis is presented in Figure 3(b). Consider the original discrete reasoning state space $\mathcal{S}$ and its corresponding continuous manifold $\tilde{\mathcal{S}}$, where $\mathcal{S} \subset \tilde{\mathcal{S}}$. For a state transition starting from $s$, let $\vec{v}_{\text{ideal}}$ be the ideal optimization vector with domain on $\tilde{\mathcal{S}}$, and let $\vec{v}_{\text{real}}$ be the best achievable discrete optimization vector with domain on $\mathcal{S}$. Since $\vec{v}_{\text{real}}$ serves as an approximation for $\vec{v}_{\text{ideal}}$, the total vector error is $\vec{\epsilon}_{\text{total}}(s) = \vec{v}_{\text{real}} - \vec{v}_{\text{ideal}}$. We can decompose the analysis of this error into two components: the error in direction (angle) and the error in size (magnitude).

**Definition 2.11.** *(Continuum errors.) The total error for continuum approximation is characterized by two components: (1) **The angular error**, $\epsilon_A(s)$, defined as the angle between $\vec{v}_{ideal}$ and $\vec{v}_{real}$. (2) **The magnitude error**, $\epsilon_M(s)$, defined as difference in norms: $\epsilon_M(s) = |\|\vec{v}_{real}\| - \|\vec{v}_{ideal}\||$.*

We will further show the convergence relationship between the continuum errors and the density, thereby revealing the rationality of the reasoning-level state space continuous approximation.

**Theorem 2.12.** *(Convergence of the continuum error). The expected angular error $\mathbb{E}\left[\epsilon_A(s)\right]$ and expected magnitude error $\mathbb{E}\left[\epsilon_M(s)\right]$ are both upper-bounded by a function that is inversely related to the reasoning state density $\rho(K)$: $\mathbb{E}[\epsilon_A(s)] \leq \mathcal{O}\left(\rho(K)^{-\frac{1}{D}}\right)$ and $\mathbb{E}[\epsilon_M(s)] \leq \mathcal{O}\left(\rho(K)^{-\frac{1}{D}}\right)$.*

The proof is provided in Appendix B.3. Theorem 2.12 demonstrates that the total error in approximating a true gradient vector with the best possible discrete step is bounded and converges to zero as the model's expressive capacity ($K$) increases. By ensuring that for a sufficiently $K$, a discrete step vector $\vec{v}_{\text{real}}$ that is an arbitrarily accurate approximation of the ideal vector $\vec{v}_{\text{ideal}}$ can always be found.

With the continuum nature of the reasoning space established, we bridge the gap between discrete language generation and classical learning theories. This continuum approximation is not merely a conceptual shift, it is the critical step that licenses the use of powerful analytical tools. Specifically, it allows us to conceptualize a smooth reasoning loss landscape and thus define a corresponding gradient, enabling us to model the generation of a new reasoning step as a descent-like update.

Building upon this, we can frame the entire reasoning process as a trajectory in a continuous state space, and link many LLM phenomena, such as hallucination, prompt sensitivity, emergent abilities, and effectiveness of external slow-thinking strategies, with classical theoretical results. A brief discussion is provided in Appendix E. In the next section, we take the "overthinking" as a detailed example to reveal the potential of CoT-Space for revitalizing the classical theoretical results.

## 3 ANALYTICAL APPLICATIONS FOR OVERTHINKING WITHIN CoT-SPACE

This section utilizes the CoT-Space framework and combines classic learning theories to analyze the "overthinking" phenomenon, thus yielding the potential of CoT-Space. We provide a theoretical foundation by examining the issue through two lenses: optimization noise (Subsection 3.1) and generalization risk (Subsection 3.2), followed by a summary of our theoretical insights (Subsection 3.3).

### 3.1 REASONING AS OPTIMIZATION: A PERSPECTIVE OF NOISE

Leveraging the continuous nature of CoT-Space, we frame LLM reasoning as an optimization process analogous to ML, as shown in Figure 4. In this analogy, an LLM policy iteratively refines its reasoning state towards a solution, much like an optimizer updates parameters. This perspective allows us to apply classical theories like sharpness-aware minimization, which suggests that an optimal amount of noise promotes convergence to better solutions. The CoT length functions as a learning rate that controls optimization noise: if too short, the process fails to converge; if too long, it leads to "overthinking", akin to finding a sharp, poor-quality minimum.

Building on this, we can model the optimization of LLM reasoning as a form of reasoning loss descent and perform a theoretical analysis of the relationship between the noise scale and CoT length. We model the reasoning process as a Stochastic Differential Equation (SDE) on the continuous manifold (details in Appendix B.4). In this model, $g$ represents the noise scale of the optimization trajectory.

**Theorem 3.1.** *(Relationship between noise scale and CoT length.) In a given optimization space, the noise scale $g$ is inversely proportional to the reasoning length $L$, i.e. $g \propto \frac{1}{L}$.*

The proof is provided in Appendix B.4. Theorem 3.1 clarifies the relationship between the noise $g$ and the CoT length $L$ in the same reasoning process. According to the sharpness-aware minimization

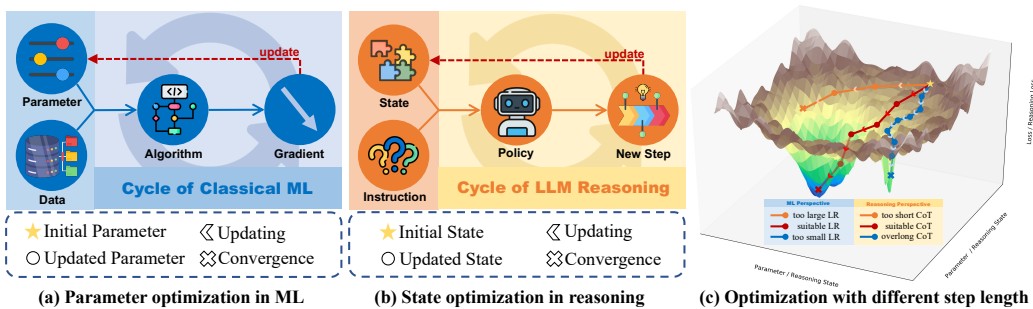

(a) Parameter optimization in ML  (b) State optimization in reasoning  (c) Optimization with different step length

Figure 4: **Modeling LLM reasoning as an optimization process.** (a) Traditional ML performs parameter optimization by updating model weights. (b) Our framework models reasoning as state optimization, where new reasoning steps iteratively update the current state. (c) This panel visualizes this process on a reasoning loss landscape, where the CoT length is analogous to a learning rate; an optimal value is crucial for converging to a high-quality, generalizable solution.

theorems, given a reasoning task, there exists an optimal noise scale $g_{\text{opt}}$ in the reasoning process, which corresponds to an optimal CoT length $L_{\text{opt}} \propto \frac{1}{g_{\text{opt}}}$. In essence, this theoretical analysis provides a fundamental basis for why an optimal CoT length is crucial for effective reasoning, highlighting its role in introducing the right amount of noise to achieve a flat, high-quality, and generalizable solution.

We note that while real-world optimization dynamics involve complex, state-dependent noise, modeling $\eta(t)$ as white noise serves as a necessary first-order approximation. This simplification allows us to isolate the fundamental inverse relationship between CoT length and the effective noise scale ($g \propto 1/L$) without the obfuscation of higher-order interference. This relationship holds conceptually: longer reasoning sequences naturally integrate more variance, necessitating a lower noise scale per step to maintain stability.

## 3.2 REASONING AND GENERALIZATION: A PERSPECTIVE OF RISK

The previous section considered reasoning from the perspective of noise. Here, we analyze the generalization risk of a policy to provide further theoretical insights, focusing on the upper bounds of reasoning risks. In the context of CoT-Space, risks can be defined similarly to classical ML. The total expected error of a policy $\pi$ on the true data distribution $\mathcal{D}$, denoted by $R(\pi) = \mathbb{E}_{q \sim \mathcal{D}}[C(s_q^\pi)]$, can be decomposed into two distinct components: $R(\pi) = \hat{R}_n(\pi) + \left(R(\pi) - \hat{R}_n(\pi)\right)$. Here, $\hat{R}_n(\pi) = \frac{1}{n} \sum_{i=1}^n C(s_{q_i}^\pi)$ is the empirical loss (a.k.a. bias) on the training set. $R(\pi) - \hat{R}_n(\pi) = \mathbb{E}_{q \sim \mathcal{D}}[C(s_q^\pi)] - \frac{1}{n} \sum_{i=1}^n C(s_{q_i}^\pi)$ is the reasoning generalization error (a.k.a. virance). In the following parts, we will analyze how each component behaves as a function of the CoT complexity.

### 3.2.1 AN UPPER BOUND FOR GENERALIZATION ERROR: THE OVERFITTING RISK

Given $n$ queries $\{q_i\}_{i=1}^n$ sampled i.i.d. from a distribution $\mathcal{D}$, and a policy $\pi$ learned from $\{q_i\}_{i=1}^n$, let's denote $s_q^\pi = \pi(q)$ as the end reasoning state output by $\pi$ on query $q$. The reasoning generalization error of $\pi$ on $\mathcal{D}$ is defined as: $R(\pi) - \hat{R}_n(\pi) = \mathbb{E}_{q \sim \mathcal{D}}[C(s_q^\pi)] - \frac{1}{n} \sum_{i=1}^n C(s_{q_i}^\pi)$. This error quantifies a policy's ability to generalize its reasoning to unseen queries, a concept parallel to the generalization error in traditional machine learning theory. By applying classical PAC-Bayesian and information-theoretic results of ML, we can derive an upper bound for the reasoning generalization error.

**Theorem 3.2.** *(Information-theoretical generalization upper bound for LLM reasoning.) With CoT-Space defined above, if the policy $\pi$ is trained on an i.i.d. drawn training set $S$ with size $n$, the following upper bound for the reasoning generalization error holds:*

$$|\mathbb{E}[R(\pi) - \hat{R}_n(\pi)]| \leq \sqrt{\frac{C_{max}^2 \cdot (\mathbb{E}_{\pi,S}[K]) \cdot \log |\mathcal{A}|}{2n}},$$

*where $C_{max}$ is the maximum possible value of the reasoning loss $C(\cdot)$, $L$ is the number of reasoning steps in a CoT data, $|\xi_i|$ is the number of tokens in the $i$-th reasoning step in a CoT data, $\mathbb{E}[K] \approx \mathbb{E}[L] \cdot \mathbb{E}[|\xi|]$ is the expected total number of tokens in the generated CoT.*

A detailed discussion and proof are provided in Appendix B.5. Theorem 3.2 explicitly connects the generalization error to the observable, structural properties of the reasoning process. It suggests that overfitting can occur in overlong response length. This provides a direct theoretical basis for regularizing the CoT generation process in RL to improve reasoning generalization.

It is important to acknowledge that standard generalization bounds typically assume i.i.d. data distributions, whereas RL involves non-stationary, on-policy data generation. However, we apply these classical bounds here as a conceptual proxy to capture the structural relationship between policy complexity (proxied by CoT length $L$) and generalization risk. This theoretical framing allows us to mechanistically interpret "overthinking" as a form of overfitting, where the policy memorizes complex reasoning paths rather than learning generalizable logic. Developing tighter bounds specifically for the non-stationary RL setting remains a critical direction for future work.

### 3.2.2 A LOWER BOUND FOR EMPIRICAL LOSS: THE UNDERFITTING RISK

The empirical loss, $\hat{R}_n(\pi)$, measures a policy's ability to solve the problems it has already encountered. For complex reasoning tasks, a certain minimum number of steps is required for a correct solution. Intuitively, if $L$ is too small, the policy lacks the necessary "thinking space" to solve the queries, even those in the training set. This leads to high empirical loss and underfitting. As $L$ increases, the policy has more capacity to find the correct reasoning path, so the empirical loss $\hat{R}_n(\pi)$ generally decreases.

Building upon this intuition, we can derive a lower bound for the empirical reasoning loss $\hat{R}_n(\pi)$.

**Theorem 3.3.** *With CoT-Space defined above, if the policy $\pi$ is trained on an i.i.d. drawn training set, the following lower bound for the empirical reasoning risk holds:*

$$\hat{R}_n(\pi) \geq P_\pi(L < L^*) \cdot C_{fail},$$

*where $L^*$ is the intrinsic required reasoning depth for the query. Note that $L^*$ is a theoretical property of the problem difficulty and is unobservable in practice. This theorem formalizes the intuition that if a policy generates a CoT shallower ($L$) than the problem requires ($L^*$), it is guaranteed to fail (underfit). $C_{fail}$ is the upper bound of the reasoning loss in the failed reasoning results.*

A detailed proof is provided in Appendix B.6. Theorem 3.3 provides a formal basis for the "underfitting" portion of our trade-off curve. It demonstrates that the empirical loss is directly constrained by the probability that the policy generates a CoT that is too short for the problem's intrinsic complexity. A policy that tends to produce short CoTs will have a high failure rate on non-trivial problems. Consequently, its empirical loss $\hat{R}_n(\pi)$ is guaranteed to be high. This formalizes why a policy must maintain enough CoT length $L$ to achieve a low empirical loss, thus completing our explanation for the tradeoff in our total error analysis and the existence of an optimal $L_{\text{opt}}$.

### 3.3 TAKEAWAYS

Our previous analysis from the perspectives of noise (Subsection 3.1) and risk (Subsection 3.2) yields the following key conclusions about the factors governing the convergence of the optimal CoT length, $L_{\text{opt}}$, in LLM reasoning via RL.

**Remark 1.** The intrinsic difficulty of a task dictates $L_{\text{opt}}$. Per Theorem 3.3, more complex tasks require a greater reasoning depth ($L^*$) to avoid underfitting. Consequently, an effective policy must learn a sufficiently long $L_{\text{opt}}$ commensurate with the task's complexity to minimize empirical loss.

**Remark 2.** Model capacity is inversely related to the optimal CoT length due to overfitting risk. According to Theorem 3.2, higher-capacity models face a greater generalization error, which scales with CoT length $L$. To mitigate this risk, more powerful models learn to produce more concise reasoning, converging to a shorter $L_{\text{opt}}$ to regularize their outputs.

**Remark 3.** The converged CoT length is independent of the specific RL algorithm used. Our analysis indicates that $L_{\text{opt}}$ is an intrinsic property determined by task difficulty and model capacity. The algorithm's role is more likely to effectively optimize the policy to find this good solution.

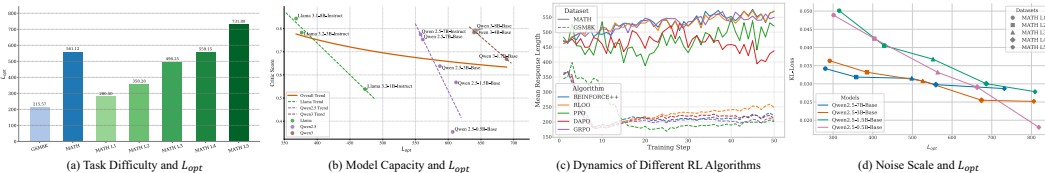

(a) Task Difficulty and $L_{opt}$ (b) Model Capacity and $L_{opt}$ (c) Dynamics of Different RL Algorithms (d) Noise Scale and $L_{opt}$

Figure 5: **Empirical validation of factors governing optimal CoT length ($L_{\mathbf{opt}}$).** (a) **Task difficulty:** $L_{opt}$ increases with task complexity. (b) **Model capacity:** More powerful models converge to a shorter $L_{opt}$ to mitigate overfitting. (c) **RL algorithm:** The converged $L_{opt}$ is largely algorithm-agnostic. (d) **Optimization noise:** $L_{opt}$ is inversely correlated with noise (proxied by KL-Loss).

**Remark 4.** The optimization noise scale is inversely correlated with the converged CoT length. Theorem 3.1 shows that higher noise levels incentivize convergence to a shorter $L_{opt}$, as noisier environments favor more robust, concise reasoning paths that are less susceptible to disruption.

To validate these theoretical claims, the following section presents a series of experiments designed to systematically test each of these four remarks.

## 4 EXPERIMENTS

This section presents a series of experiments designed to empirically validate the theoretical insights established in the preceding sections. We systematically investigate the relationship between the optimal CoT length ($L_{opt}$) and key factors.

We evaluate on the MATH (Hendrycks et al., 2021) and GSM8K (Cobbe et al., 2021) benchmarks. The MATH training set is subdivided into five difficulty levels (Level 1–5). We utilize a diverse suite of base models including the Qwen2.5 series (Qwen, 2025a), Qwen3 series (Qwen, 2025b), and Llama3 series (Meta, 2024). We compare five mainstream RL algorithms: GRPO (Shao et al., 2024), PPO (Schulman et al., 2017), DAPO (Yu et al., 2025), Reinforce++ (Hu et al., 2025), and RLOO (Ahmadian et al., 2024). The detailed experimental setup is listed in Appendix C.

**Task Difficulty and $L_{\mathbf{opt}}$.** To validate Remark 1, we test on tasks of increasing difficulty (GSM8K and five levels of MATH) with Qwen2.5-7B-Base. As shown in Figure 5(a), $L_{opt}$ demonstrates a clear monotonic increase with task complexity. This aligns with Theorem 3.3, as more challenging tasks require a greater reasoning depth ($L^*$). To avoid the high empirical loss from underfitting, the policy learns to generate a correspondingly longer CoT. Additional validations are in Appendix D.

**Model Capacity and $L_{\mathbf{opt}}$.** To verify Remark 2, we examine models with varying capacities. Figure 5(b) reveals a distinct negative correlation: within each model family (e.g., Llama 3, Qwen 2.5, and Qwen 3), larger and higher-performing models consistently converge to a shorter $L_{opt}$. This finding supports our analysis based on Theorem 3.2. Larger models are more prone to overfitting, a risk exacerbated by longer CoTs. Consequently, they learn to produce more concise reasoning to minimize total error by balancing empirical performance with generalization.

**RL Algorithm and $L_{\mathbf{opt}}$.** To test Remark 3, we compare five different RL algorithms. As shown in Figure 5(c), despite different learning dynamics, all algorithms guide the policy to a similar final $L_{opt}$ for a given task and model, even though the initial dynamics differ. This supports our claim that the RL algorithm primarily facilitates optimization, while the final $L_{opt}$ is an intrinsic property determined by task difficulty and model capacity, not the specific algorithm used.

**Noise Scale and $L_{\mathbf{opt}}$.** Finally, to validate Remark 4, we use the KL-Loss as a proxy for optimization noise. In RL algorithms like PPO, the KL divergence term penalizes deviations from the reference policy, effectively regulating the exploration range. A higher KL-Loss indicates a larger deviation and higher variance in the policy's update trajectory, analogous to a higher noise scale in our theoretical model. Figure 5(d) shows a clear inverse correlation between the final KL-Loss and $L_{opt}$. This result substantiates Theorem 3.1, confirming that noisier optimization environments favor shorter CoTs to ensure robust convergence, as longer reasoning paths are more susceptible to disruption.

## 5 RELATED WORK

**Reasoning with LLMs.** Prompting techniques have advanced LLM reasoning beyond few-shot learning (Brown et al., 2020). Chain-of-Thought (CoT) prompting elicits intermediate reasoning steps (Wei et al., 2022b), with methods like Self-Consistency improving robustness (Wang et al., 2022). Other reasoning formats include Program-of-Thought (Chen et al., 2022) and internal thought tokens (Zelikman et al., 2024; Chen et al., 2025a). While theoretical analyses are emerging (Ton et al., 2024), much of this work remains empirical and lacks a unifying theoretical foundation.

**RL for LLM Reasoning.** Reinforcement Learning (RL) is a key method for test-time scaling, a technique proven to enhance LLM reasoning (Snell et al., 2024; Wu et al., 2024; Chen et al., 2025b). Research in this area follows two main tracks: developing new post-training frameworks like VC-PPO, DAPO, and VAPO (Yuan et al., 2025; Yu et al., 2025; Yue et al., 2025b), and understanding the fundamental effects of RL. Studies in the latter track investigate RL's advantages over SFT (Setlur et al., 2025), its impact on knowledge boundaries (Yue et al., 2025a), the role of reward modeling (Shao et al., 2025), and its tendency to amplify pre-trained behaviors (Zhao et al., 2025). The absence of a common theoretical framework, however, can lead to divergent findings.

**Optimization and Generalization.** A core concept in machine learning optimization is that flatter minima in the loss landscape often lead to better generalization, a principle central to Sharpness-Aware Minimization (SAM) (Keskar et al., 2016). This effect is closely linked to the noise inherent in the optimization process (Smith & Le, 2018). Generalization itself, which measures performance on unseen data, is often analyzed through the powerful lens of information theory. Concepts like entropy and mutual information are used to establish theoretical bounds on the generalization capacity of deep learning models (Russo & Zou, 2019; Xu & Raginsky, 2017) and have recently been applied to analyze synthetic data generation and reasoning errors in LLMs (Gan & Liu, 2024; Ton et al., 2024).

## 6 CONCLUSION

In this paper, we introduced CoT-Space, a novel theoretical framework that addresses the critical misalignment between token-level RL theories and the reasoning-level nature of CoT by reframing LLM reasoning as a continuous optimization problem. This shift in perspective serves as a conceptual bridge, demonstrating that the foundational principles of classical machine learning are not obsolete but can be powerfully revitalized to analyze modern LLM behaviors. By revitalizing classical concepts of optimization and risk analysis, we demonstrate that a fundamental trade-off between underfitting (shorter CoTs) and overfitting (longer CoTs) governs the reasoning process. This analysis provides a solid theoretical grounding for the existence of an optimal CoT length, $L_{opt}$, that minimizes total error. Looking forward, the CoT-Space framework moves beyond description to offer a prescriptive foundation for future algorithms. By establishing the continuum nature of reasoning, our work suggests two promising directions: (1) Latent Space Optimization, where reasoning is performed directly within the continuous manifold via auto-encoders rather than discrete token generation, and (2) Continuous Loss Design, motivating the development of dense verifiers that output smooth "distance-to-solution" signals rather than sparse binary rewards. We elaborate on these implications in Appendix E.4.

## ETHICS STATEMENT

The authors have read and adhered to the ICLR Code of Ethics. This work is primarily theoretical, aiming to establish a foundational understanding of the reasoning mechanisms in Large Language Models trained with Reinforcement Learning. Our research does not involve human subjects or the collection of new sensitive data. There may be some potential societal consequences of our work, none of which we feel must be specifically highlighted here.

## REPRODUCIBILITY STATEMENT

To ensure the reproducibility of our findings, we have made our resources publicly available. For the theoretical contributions, we provide complete mathematical derivations and proofs for all theorems

and lemmas in Appendix B. For empirical results, all code used for the experiments presented in this paper is open-sourced and can be accessed through the anonymous GitHub repository linked in the abstract. A comprehensive description of the experimental setup is detailed in Appendix C. We believe these resources provide the necessary details for the research community to verify our claims and build upon our work.

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

## A USE OF LARGE LANGUAGE MODELS

In line with the ICLR policy, we disclose that Large Language Models (LLMs) were used as a general-purpose writing assistant during the preparation of this manuscript. The primary role of LLMs was to aid in polishing the text, which included improving grammar, refining sentence structure for clarity, and checking for stylistic consistency.

## B PROOFS

### B.1 PROOF OF THEOREM 2.10

*Proof.* We begin with the definition of the number of realizations for a complete reasoning process.

**Definition B.1.** *(Number of realizations.) For a complete reasoning trajectory of $L$ steps, $(\xi_1, \xi_2, ..., \xi_L)$, its total number of token-level realizations, $N_{realize}$, is the size of the Cartesian product of the semantic equivalence sets for each step.*

$$N_{realize}(\xi_1, ..., \xi_L) = \prod_{l=1}^{L} |\mathcal{V}(\xi_l, k_l)|.$$

Consider a single, fixed reasoning-level path $(\xi_1, ..., \xi_L)$, where each step $\xi_l$ is realized by an average of $k$ tokens. By Assumption 2.9, the number of token-level realizations for this single path, $N_{\text{realize}}^{K}$, is bounded below:

$$N_{\text{realize}}^{K} \geq \prod_{l=1}^{L} c^{k_l} = c^{\sum_{l=1}^{L} k_l} = c^{kL} = c^{K}. \tag{1}$$

This shows that the number of token-level trajectories corresponding to even a single reasoning path grows exponentially with the total number of tokens, denoted as $K = kL$.

Next, we consider the total number of distinct, valid reasoning states, $|\mathcal{S}_{\text{reasoning}}^{(K)}|$. For complex problems, given a total token amount $K$, the valid reasoning paths can be realized in distinct lengths that are no longer than $K$, which means $|\mathcal{S}_{\text{reasoning}}^{(K)}| = \sum_{K' \leq K} N_{\text{realize}}^{K'}$. According to Equation (1), the number of realizations is bounded by an exponential term of the total number of tokens $K$, thus $|\mathcal{S}_{\text{reasoning}}^{(K)}|$ is of the same order with $N_{\text{realize}}^{K}$.

$$|\mathcal{S}_{\text{reasoning}}^{(K)}| = \Theta(c^{K}). \tag{2}$$

From Definition 2.8, the reasoning state density $\rho(K)$ is thus:

$$\rho(K) = \frac{|\mathcal{S}_{\text{reasoning}}^{(K)}|}{|\mathcal{V}_{\text{semantic}}|} = \Theta(c^{K}). \tag{3}$$

Assuming the semantic volume $\mathcal{V}_{\text{semantic}}$ for a given problem domain is fixed, the state density $\rho(K)$ thus grows exponentially with $K$.

Finally, we connect high density to the expected distance between adjacent points with the following lemma.

**Lemma B.2.** *(Nearest neighbor distance). Let points be distributed in a $d$-dimensional Euclidean space $\mathbb{R}^d$ according to a homogeneous Poisson point process with density $\rho$. The expected distance to the $k$-th nearest neighbor, denoted by $\mathbb{E}[R_k]$, is given by*

$$\mathbb{E}[R_k] = \frac{1}{(k-1)!} \left(\frac{1}{\rho C_d}\right)^{1/d} \Gamma\left(k + \frac{1}{d}\right),$$

*where $C_d = \frac{\pi^{d/2}}{\Gamma(\frac{d}{2}+1)}$ is the volume of a unit $d$-dimensional ball and $\Gamma(\cdot)$ is the Gamma function.*

The proof is provided in Appendix B.2. Lemma B.2 implies the relation between the density and the expected distance of the points in a space. By only considering the nearest adjacent points, in a $D$-dimensional space, the expected distance $\mathbb{E}[\text{dist}_{s_i \bowtie s_j}(s_i, s_j)]$ between adjacent nearest neighbor points $s_i, s_j$ is inversely related to the density, approximately scaling as:

$$\mathbb{E}_{s_i \bowtie s_j}[\text{dist}(s_i, s_j)] \propto \rho^{-1/D}. \tag{4}$$

Since $\rho(K)$ grows exponentially, it follows that:

$$\lim_{K \to \infty} \mathbb{E}_{s_i \bowtie s_j}[\text{dist}(s_i, s_j)] = 0. \tag{5}$$

When the expected distance between states approaches zero, it implies that for any state and any direction, another valid state can be found within an infinitesimally small neighborhood. This is the defining characteristic of a continuous space or manifold. Therefore, as the reasoning token amount $K$ increases, the discrete CoT-Space increasingly resembles a continuum, which validates the application of continuous optimization tools for its analysis.

This finishes the proof. $\qquad\qquad\square$

### B.2 PROOF OF LEMMA B.2

*Proof.* The proof is built upon Haenggi (2005) and Miyagawa (2018).

First, we derive the probability density function, $f_k(r)$, for the distance to the $k$-th nearest neighbor. The cumulative distribution function (CDF), $F_k(r)$, is the probability that a $d$-dimensional ball $B_r$ of radius $r$ contains at least $k$ points. The volume of this ball is $V_d(r) = C_d r^d$.

The number of points $X$ in a region of volume $V$ follows a Poisson distribution with mean $\rho V$:

$$P(X = x) = \frac{(\rho V)^x}{x!} e^{-\rho V}. \tag{6}$$

The CDF is therefore 1 minus the probability of finding fewer than $k$ points in the ball $B_r$:

$$F_k(r) = P(\text{points in } B_r \geq k) = 1 - \sum_{x=0}^{k-1} P(X = x) = 1 - e^{-\rho C_d r^d} \sum_{x=0}^{k-1} \frac{(\rho C_d r^d)^x}{x!}. \tag{7}$$

The PDF is the derivative of the CDF with respect to $r$:

$$f_k(r) = \frac{\mathrm{d}}{\mathrm{d}r} F_k(r) = -\frac{\mathrm{d}}{\mathrm{d}r} \left( e^{-\rho C_d r^d} \sum_{x=0}^{k-1} \frac{(\rho C_d r^d)^x}{x!} \right). \tag{8}$$

Using the product rule, the derivative simplifies due to a telescoping sum, yielding:

$$f_k(r) = \frac{d(\rho C_d)^k r^{dk-1}}{(k-1)!} e^{-\rho C_d r^d}. \tag{9}$$

The expected value $\mathbb{E}[R_k]$ is defined as the integral of $r$ times its PDF over its domain:

$$\mathbb{E}[R_k] = \int_0^\infty r \cdot f_k(r) \, \mathrm{d}r. \tag{10}$$

Substituting the PDF we derived:

$$\mathbb{E}[R_k] = \int_0^\infty r \cdot \left( \frac{d(\rho C_d)^k r^{dk-1}}{(k-1)!} e^{-\rho C_d r^d} \right) \mathrm{d}r. \tag{11}$$

We combine the terms involving $r$ and move the constants outside the integral:

$$\mathbb{E}[R_k] = \frac{d(\rho C_d)^k}{(k-1)!} \int_0^\infty r^{dk} e^{-\rho C_d r^d} \, \mathrm{d}r. \tag{12}$$

To solve the integral, we perform a change of variables. Let $t = \rho C_d r^d$. This implies:

$$r = \left( \frac{t}{\rho C_d} \right)^{1/d} \quad \text{and} \quad dt = d\rho C_d r^{d-1} \, dr. \tag{13}$$

The integral becomes:

$$
\begin{aligned}
\int_0^\infty r^{dk} e^{-\rho C_d r^d} \, dr &= \int_0^\infty \left( \frac{t}{\rho C_d} \right)^k e^{-t} \frac{dt}{d\rho C_d r^{d-1}} \\
&= \int_0^\infty \left( \frac{t}{\rho C_d} \right)^k e^{-t} \frac{dt}{d\rho C_d (t/(\rho C_d))^{(d-1)/d}} \\
&= \frac{1}{d(\rho C_d)^{k+1-(d-1)/d}} \int_0^\infty t^{k-(d-1)/d} e^{-t} \, dt \\
&= \frac{1}{d(\rho C_d)^{k+1/d}} \int_0^\infty t^{(k+1/d)-1} e^{-t} \, dt.
\end{aligned}
\tag{14}
$$

The remaining integral is the definition of the Gamma function, $\Gamma(k + 1/d)$. Thus:

$$\int_0^\infty r^{dk} e^{-\rho C_d r^d} \, dr = \frac{\Gamma(k+1/d)}{d(\rho C_d)^{k+1/d}}. \tag{15}$$

Substituting this result back into the expression for $\mathbb{E}[R_k]$:

$$\mathbb{E}[R_k] = \frac{d(\rho C_d)^k}{(k-1)!} \left( \frac{\Gamma(k+1/d)}{d(\rho C_d)^{k+1/d}} \right). \tag{16}$$

The terms $d$ and $(\rho C_d)^k$ cancel out, leaving:

$$\mathbb{E}[R_k] = \frac{1}{(k-1)!} \frac{\Gamma(k+1/d)}{(\rho C_d)^{1/d}}. \tag{17}$$

Which can be written as the final result:

$$\mathbb{E}[R_k] = \frac{1}{(k-1)!} \left( \frac{1}{\rho C_d} \right)^{1/d} \Gamma \left( k + \frac{1}{d} \right). \tag{18}$$

This completes the proof. $\qquad\qquad\qquad\qquad\qquad\qquad\qquad\qquad\qquad\qquad\square$

### B.3 PROOF OF THEOREM 2.12

Our goal is to show that both error components converge to zero as the reasoning state density $\rho(K)$ increases. Consider a $D$-dimensional semantic space. This analysis is predicated on a reasonable assumption about the locality of a single reasoning step.

**Assumption B.3.** *(Bounded step optimization). Any single reasoning step corresponds to a transition in the semantic space with an effective maximum radius $R_{max}$ and a minimum radius $R_{min}$. Thus, from a state $s$, the set of all possible subsequent states is contained within the region between two concentric $D$-dimensional hyperspheres centered at $s$ with radii $R_{min}$ and $R_{max}$. This region can be denoted as $B_R(s)$.*

Following Assumption B.3, we continue to use the following definition to characterize the distribution structure of state points in the semantic space.

**Definition B.4.** *(Maximal void.) The maximal void $B_R^{hole}(s)$ is defined as the largest void hypersphere region within the bounded step radius. Formally, it is the largest hypersphere subspace of $B_R(s)$ that does not contain any state points:*

$$B_R^{hole}(s) = \arg \max_B \{ r(B) \mid B \subseteq B_R(s) \text{ and } B \cap \mathcal{S} = \emptyset \},$$

*where $r(\cdot)$ is the radius of a ball and $\mathcal{S}$ is the reasoning state space.*

Let $R_{\text{hole}}$ be the radius of $B_R^{\text{hole}}(s)$. We further show the relationship between $R_{\text{hole}}$ and the reasoning state density $\rho$.

**Lemma B.5.** *(Maximal void radius and reasoning state density.) In a $D$-dimensional semantic space, the radius of the maximal void $R_{hole}$ and the density of reasoning state points $\rho(K)$ have the following relationship:*

$$R_{hole} = \mathcal{O}\left(\rho(K)^{-\frac{1}{D}}\right).$$

*Proof.* The proof relies on the connection between the average nearest-neighbor distance and the scale of the largest voids in a spatial point process.

As established in Theorem 2.10, which builds upon the theory of spatial point processes (Lemma B.2), the expected distance between adjacent reasoning states in a $D$-dimensional semantic space is inversely related to the state density $\rho(K)$:

$$\mathbb{E}_{s_i \bowtie s_j}[\text{dist}(s_i, s_j)] \propto \rho(K)^{-1/D}. \tag{19}$$

The radius of the maximal void, $R_{\text{hole}}$, represents the largest possible distance from a point within that void to the nearest available reasoning state. It can thus be understood as a measure of the worst-case nearest-neighbor distance in a local region.

In the study of stochastic geometry, for a homogeneous point process, the distribution of nearest-neighbor distances is concentrated around its mean. Consequently, the scaling of the maximal values is of the same order as the scaling of the expected value.

Given that the expected distance is proportional to $\rho(K)^{-1/D}$, it follows that the maximal void radius, representing the worst-case local distance, must adhere to the same scaling law. Therefore, we can conclude that:

$$R_{\text{hole}} = \mathcal{O}\left(\rho(K)^{-\frac{1}{D}}\right). \tag{20}$$

This completes the proof. $\square$

With Lemma B.5, we then commence to analyze the convergence of the two continuum errors, respectively.

**Analysis of the Angular Error $\epsilon_A(s)$**

The angular error measures the difference in direction between discrete steps and continuous steps. Building upon the previous discussion, we can bound the angular error by the following result.

**Lemma B.6.** *(Convergence of the Angular Error). The expected angular error $\mathbb{E}\left[\epsilon_A(s)\right]$ is bounded above by a function that is inversely related to the reasoning state density $\rho(K)$:*

$$\mathbb{E}[\epsilon_A(s)] \leq \mathcal{O}\left(\rho(K)^{-\frac{1}{D}}\right).$$

*Proof.* The angular error $\epsilon_A(s)$ is defined as the angle between the ideal step vector $\vec{v}_{\text{ideal}}$ and the best achievable discrete step vector $\vec{v}_{\text{real}}$. This error is determined by the directional sparsity of available states in the semantic space.

The worst-case error occurs when the ideal vector $\vec{v}_{\text{ideal}}$ points directly towards the center of the maximal void, $B_R^{\text{hole}}(s)$, within the local neighborhood $B_R(s)$. Let the radius of this maximal void be $R_{\text{hole}}$. The ideal target state $p_{\text{ideal}} = s + \vec{v}_{\text{ideal}}$ is at the center of this void.

By definition, the best achievable subsequent state, $s_j^*$, must lie on or outside the boundary of this void. The vector difference $s_j^* - p_{\text{ideal}}$ represents the deviation from the ideal target.

Let's consider the triangle formed by the points $s$ (current state), $p_{\text{ideal}}$ (ideal target), and $s_j^*$ (best achievable state). The angular error $\epsilon_A(s)$ is the angle at vertex $s$. Using the small-angle approximation for sine, which is valid as the space becomes dense, we can bound the angle:

$$\epsilon_A(s) \approx \sin(\epsilon_A(s)) \approx \frac{\|s_j^* - p_{\text{ideal}}\|}{\|\vec{v}_{\text{real}}\|} \leq \frac{\|s_j^* - p_{\text{ideal}}\|}{R_{\text{min}}}. \tag{21}$$

In the worst-case scenario, the distance from the ideal point to the best real point $\|s_j^* - p_{\text{ideal}}\|$ is bounded by the properties of the void. This distance is on the order of the void's radius, $R_{\text{hole}}$. Since the step size $\|\vec{v}_{\text{real}}\|$ is bounded and non-zero, the angular error is directly proportional to the radius of the maximal void:

$$\epsilon_A(s) = \mathcal{O}(R_{\text{hole}}). \tag{22}$$

Taking the expectation over all possible configurations of the state space, we have:

$$\mathbb{E}[\epsilon_A(s)] = \mathcal{O}(\mathbb{E}[R_{\text{hole}}]). \tag{23}$$

By combining this with Lemma B.5, we establish the bound on the expected angular error:

$$\mathbb{E}[\epsilon_A(s)] \leq \mathcal{O}(\rho(K)^{-\frac{1}{D}}). \tag{24}$$

As established in Theorem 2.10, the reasoning state density $\rho(K)$ grows exponentially with the total reasoning token amount $K$. Consequently, as $K \to \infty$, $\rho(K) \to \infty$, and thus $\lim_{K \to \infty} \mathbb{E}[\epsilon_A(s)] = 0$. This demonstrates that the direction of the best discrete step converges to the direction of the ideal continuous gradient.

This completes the proof.

$\square$

**Analysis of the Magnitude Error** $\epsilon_M(s)$

The magnitude error quantifies how well the length of the discrete step matches the length of the ideal step. This insight provides a direct way to bound this error.

**Lemma B.7.** *(Convergence of the Magnitude Error). The expected magnitude error $\mathbb{E}[\epsilon_M(s)]$ is also bounded above by a function that is inversely related to the reasoning state density $\rho(K)$:*

$$\mathbb{E}[\epsilon_M(s)] \leq \mathcal{O}\left(\rho(K)^{-\frac{1}{D}}\right).$$

*Proof.* By the reverse triangle inequality, the magnitude error is bounded by the norm of the vector difference:

$$\epsilon_M(s) = |\|\vec{v}_{\text{real}}\| - \|\vec{v}_{\text{ideal}}\|| \leq \|\vec{v}_{\text{real}} - \vec{v}_{\text{ideal}}\|. \tag{25}$$

Let $p_{\text{ideal}} = s + \vec{v}_{\text{ideal}}$ be the ideal target state. The best achievable state $s_j^*$ is the valid state closest to $p_{\text{ideal}}$. The vector difference is then $\|(s_j^* - s) - (p_{\text{ideal}} - s)\| = \|s_j^* - p_{\text{ideal}}\|$.

In the worst-case scenario, the ideal target $p_{\text{ideal}}$ falls within the largest void. The closest available state $s_j^*$ must lie on or outside the boundary of this void. The distance between any point inside a hypersphere and its boundary is at most its diameter, $2R_{\text{hole}}$. Therefore, the distance between the ideal target and the best real state is bounded:

$$\|s_j^* - p_{\text{ideal}}\| \leq 2R_{\text{hole}}. \tag{26}$$

Combining these steps, we get a bound for the magnitude error:

$$\epsilon_M(s) \leq 2R_{\text{hole}}. \tag{27}$$

Taking the expectation and applying Lemma B.5, we find that the expected magnitude error is bounded by the same scaling law as the angular error:

$$\mathbb{E}[\epsilon_M(s)] \leq 2\mathbb{E}[R_{\text{hole}}] = \mathcal{O}\left(\rho(K)^{-1/D}\right). \tag{28}$$

As $K \to \infty$, $\rho(K) \to \infty$, and thus $\lim_{K \to \infty} \mathbb{E}[\epsilon_M(s)] = 0$.

This completes the proof. $\square$

### B.4 PROOF OF THEOREM 3.1

*Proof.* As previously established, when viewed from a reasoning-level perspective, the reasoning space can be considered approximately continuous, i.e., $s_o \in \mathcal{S}$ where $\mathcal{S}$ is a continuous space. The state is then optimized continuously based on the policy and reasoning loss. This perspective enables us to analyze the reasoning process through a loss-reduction lens.

For an $L$-step CoT reasoning process, this can be viewed as an $L$-step optimization based on the gradient of the reasoning loss. A single step of this optimization can be written as:

$$\Delta s = -\frac{1}{L} \frac{\mathrm{d}}{\mathrm{d}s} \hat{C}, \tag{29}$$

where $\hat{C}$ is a noisy estimation of the reasoning loss, influenced by the policy and random sampling. We assume that $\hat{C}$ is a random variable with expectation $C$. Here, $L$ represents the total reasoning steps, or the CoT length.

Despite the above gradient descent formulation, this optimization process can also be modeled as a stochastic differential equation (SDE) with $L$ steps:

$$\frac{\mathrm{d}s}{\mathrm{d}t} = -\frac{\mathrm{d}C}{\mathrm{d}s} + \eta(t), \tag{30}$$

where $\eta(t)$ is the noise term and $t = 1, \ldots, L$. We assume the noise term $\eta(t)$ is white noise with the statistical properties: $\langle \eta(t) \rangle = 0$ and autocorrelation $\langle \eta(t)\eta(t') \rangle = gF(s)\delta(t - t')$, where $g$ is the noise scale, $F(s)$ s a function characterizing the magnitude of the reasoning loss estimation error at state $s$.

Equations (29) and (30) formulate the optimization process of CoT reasoning from the gradient update and SDE perspective, respectively. By combining the two equations related to noise, we can thus obtain the relationship between noise scale and CoT length, as shown in Theorem 3.1. The detailed derivation of this result is provided below.

To bridge the discrete reasoning update with the continuous SDE model, we equate the variance induced by the noise term in both frameworks over a single reasoning step.

First, we analyze the noise in the discrete one-step reasoning update from Equation (29):

$$\Delta s = -\frac{1}{L} \left( \frac{dC}{ds} + \left( \frac{d\hat{C}}{ds} - \frac{dC}{ds} \right) \right). \tag{31}$$

The noise originates from the estimation error, which we define as $\alpha = \left( \frac{d\hat{C}}{ds} - \frac{dC}{ds} \right)$. We assume this error has a variance $\langle \alpha^2 \rangle = F(s)$, where $F(s)$ is a function characterizing the magnitude of the reasoning loss estimation error at state $s$. The variance of the full noise term in the discrete update is therefore:

$$\left\langle \left( -\frac{1}{L}\alpha \right)^2 \right\rangle = \frac{1}{L^2} \langle \alpha^2 \rangle = \frac{F(s)}{L^2}. \tag{32}$$

Next, we analyze the noise from the SDE model proposed in Equation (30).

We assume the noise term $\eta(t)$ is white noise with the following statistical properties: $\langle \eta(t) \rangle = 0$ and autocorrelation $\langle \eta(t)\eta(t') \rangle = gF(s)\delta(t - t')$, where $g$ is the noise scale. To find the variance over a discrete step, we integrate the SDE over a time interval $\Delta t = 1/L$:

$$\Delta s = \int_0^{1/L} \left( -\frac{dC}{ds} + \eta(t) \right) dt = -\frac{1}{L}\frac{dC}{ds} + \int_0^{1/L} \eta(t)dt. \tag{33}$$

The variance of the noise component of this update is:

$$\left\langle \left( \int_0^{1/L} \eta(t)dt \right)^2 \right\rangle = \int_0^{1/L} \int_0^{1/L} \langle \eta(t)\eta(t') \rangle dt'dt = \int_0^{1/L} \int_0^{1/L} gF(s)\delta(t-t')dt'dt = \frac{gF(s)}{L}. \tag{34}$$

By equating the variance from the discrete update (Eq. 32) and the SDE model (Eq. 34), we can solve for the noise scale $g$:

$$\frac{F(s)}{L^2} = \frac{gF(s)}{L}. \tag{35}$$

Solving for $g$, we find:

$$g = \frac{1}{L}. \tag{36}$$

This result shows that the noise scale $g$ in our CoT-Space SDE model is inversely proportional to the CoT length $L$.

This finishes the proof. $\square$

### B.5  PROOF AND ANALYSIS OF THEOREM 3.2

To derive an upper bound for the reasoning generalization error, we can first simply adapt the PAC-Bayes framework, which provides a powerful link between generalization, policy complexity, and data volume. For our analysis within CoT-Space, we establish the following components:

- **Hypothesis Space** $\mathcal{H}$**:** The space of all possible reasoning policies, denoted by $\mathcal{H}$. Our learned policy $\pi \in \mathcal{H}$.
- **Prior Distribution** $P$**:** A distribution over the policy space $\mathcal{H}$ reflecting our initial beliefs about which policies are "good," often favoring simpler ones.
- **Posterior Distribution** $Q$**:** A distribution over $\mathcal{H}$ that is learned from the training data queries $\{q_i\}_{i=1}^n$, concentrating on policies that perform well on that data.

By applying the PAC-Bayes theorem (McAllester, 1998), we obtain a bound on the expected true loss which, when rearranged, yields an upper bound on the expected generalization error.

**Theorem B.8.** *(PAC-Bayesian generalization upper bound for LLM reasoning.) With CoT-Space defined above, the policy $\pi$ is trained on an i.i.d. drawn training set with size $n$, by assuming the reasoning loss is bounded in $[0, 1]$, for any prior $P$, with probability at least $1 - \delta$, the following holds for all posterior $Q$:*

$$\mathbb{E}_{\pi \sim Q}[R(\pi) - \hat{R}_n(\pi)] \leq \sqrt{\frac{\mathrm{KL}(Q||P) + \ln \frac{2n}{\delta}}{2n}},$$

*where $\mathrm{KL}(Q||P) = \mathbb{E}_{\pi \sim Q}[\ln \frac{Q(\pi)}{P(\pi)}]$ is the Kullback-Leibler (KL) divergence between the posterior and prior distributions.*

*Proof.* Let's denote the true reasoning loss of a policy $\pi$ as $R(\pi) = \mathbb{E}_{q \sim \mathcal{D}}[C(s_q^\pi)]$ and the empirical reasoning loss as $\hat{R}_n(\pi) = \frac{1}{n} \sum_{i=1}^n C(s_{q_i}^\pi)$. The reasoning generalization error for a specific policy $\pi$ is $R(\pi) - \hat{R}_n(\pi)$. We aim to bound this gap.

The PAC-Bayes theorem provides a bound on the expected true loss for policies drawn from the posterior $Q$. A standard version of the theorem is often presented as follows.

**Lemma B.9.** *(PAC-Bayes generalization upper bound (McAllester, 1998).) For any loss function bounded in $[0, 1]$, for any prior $P$, with probability at least $1 - \delta$ over the draw of the training set $\{q_i\}_{i=1}^n$, the following holds for all posterior distributions $Q$:*

$$\mathbb{E}_{\pi \sim Q}[R(\pi)] \leq \mathbb{E}_{\pi \sim Q}[\hat{R}_n(\pi)] + \sqrt{\frac{KL(Q||P) + \ln \frac{2n}{\delta}}{2n}},$$

*where $\mathrm{KL}(Q||P) = \mathbb{E}_{\pi \sim Q}[\ln \frac{Q(\pi)}{P(\pi)}]$ is the Kullback-Leibler (KL) divergence between the posterior and prior distributions.*

Let's adapt Lemma B.9 to our framework. We first need to normalize our reasoning loss $C(\cdot)$ to be in $[0, 1]$. Assuming there is a maximum possible loss $C_{max}$, we can define a normalized loss

$C'(\cdot) = C(\cdot)/C_{max}$. The bound then applies to the normalized losses $R'(\pi)$ and $\hat{R}'_n(\pi)$. For simplicity of notation, let's assume $C(\cdot)$ is already normalized in $[0, 1]$.

This finishes the proof. □

The bound has a clear information-theoretic meaning, which aligns with the goal of understanding reasoning phenomena:

**Complexity Term** ($\mathrm{KL}(Q\|P)$)**:** The KL divergence term measures the information gain in moving from our prior belief $P$ to our posterior belief $Q$ after seeing the data. It quantifies the complexity of the learned policy distribution. If $Q$ is very different from $P$ (i.e., $\mathrm{KL}(Q\|P)$ is large), it means we had to learn a great deal from the training data to find a good policy. This is analogous to overfitting or memorizing the training set. The bound becomes looser (larger), penalizing this complexity and suggesting a higher risk of poor generalization. If $Q$ remains close to $P$ ($\mathrm{KL}(Q\|P)$ is small), it implies the learned policy did not stray far from our initial bias towards simpler policies. The bound is tighter, suggesting better generalization.

**Data Term** ($\frac{1}{n}$)**:** The bound depends on the number of training queries $n$. As $n$ increases, the term under the square root decreases, making the bound tighter. This confirms the intuition that having more data leads to better generalization and reduces the gap between training and test performance.

In conclusion, this PAC-Bayes bound formalizes the trade-off in learning a reasoning policy. To achieve good generalization, a policy $\pi$ must not only achieve a low empirical reasoning loss $\hat{R}_n(\pi)$ on the queries it has seen, but it must also remain simple in an information-theoretic sense, by not deviating too much from a reasonable prior. This framework provides a principled way to analyze and potentially control for phenomena like "overthinking" by viewing it as a form of overfitting, which would correspond to a large $\mathrm{KL}(Q\|P)$ term.

Theorem B.8 provides a powerful, high-probability upper bound on the expected reasoning generalization error for policies sampled from the learned posterior $Q$. While this PAC-Bayesian bound is a valid, generic result, we can derive a more specific bound by using information theory, a perspective that connects the generalization error to the mutual information between the learned policy and the training data. This approach leads to the information-theoretic upper bound in Theorem 3.2.

We continue to provide the proof of Theorem 3.2 here.

*Proof.* The core idea is that a policy can only overfit to the training set $S = \{q_1, \ldots, q_n\}$ by encoding information about $S$ within its own parameters. The mutual information $I(\pi; S)$ precisely quantifies this amount of encoded information. A key result from learning theory provides the following bound on the expected generalization error:

**Lemma B.10.** *(Information-theoretical generalization upper bound (Xu & Raginsky, 2017).) With CoT-Space defined above, if the policy $\pi$ is trained on an i.i.d. drawn training set $S$ with size $n$, the following upper bound for the reasoning generalization error holds:*

$$|\mathbb{E}[R(\pi) - \hat{R}_n(\pi)]| \le \sqrt{\frac{C_{max}^2 \cdot I(\pi; S)}{2n}},$$

*where $C_{max}$ is the maximum possible value of the reasoning loss $C(\cdot)$, and the expectation is over the random draw of the training set $S$ and the (potentially stochastic) output $\pi$ of the learning algorithm.*

Lemma B.10 transforms the problem of bounding generalization error into the problem of bounding the mutual information $I(\pi; S)$. To make this bound meaningful for our framework, we must connect $I(\pi; S)$ to the structural properties of the generated Chain-of-Thought.

We then turn to bound mutual information by CoT complexity. The policy $\pi$ expresses the information it has learned from $S$ by generating a CoT, which is a sequence of reasoning steps $\xi = (\xi_1, \xi_2, \ldots, \xi_L)$. Each step $\xi_i$ is, in turn, a sequence of tokens. The total information capacity of the policy's output is therefore determined not just by the number of steps, but by the total number of tokens generated.

Let us define:

- $L$: The number of reasoning steps in a CoT.

- $|\xi_i|$: The number of tokens in the $i$-th reasoning step.

- $\sum_{i=1}^{L} |\xi_i|$: The total number of tokens in the CoT.

The mutual information $I(\pi; S)$ is upper-bounded by the entropy of the policy's output, which represents its description length or information capacity. The total number of token-level decisions the policy makes determines this capacity. Let $\mathbb{E}[L]$ be the expected number of reasoning steps (reasoning depth) and $\mathbb{E}[|\xi|]$ be the expected number of tokens per step (step verbosity). The expected total number of tokens is $\mathbb{E}[L] \cdot \mathbb{E}[|\xi|]$.

If we assume the policy generates tokens from a dictionary $\mathcal{A}$, the information required to specify one token is at most $\log |\mathcal{A}|$. Therefore, we can establish the following upper bound on the mutual information.

**Lemma B.11.** *In LLM reasoning scenarios, if the policy $\pi$ is trained on a training set $S$, the mutual information $I(\pi; S)$ is upper bounded as follows:*

$$I(\pi; S) \leq \mathbb{E}_{S,\pi} \left[ \sum_{i=1}^{L} |\xi_i| \right] \cdot \log |\mathcal{A}| = (\mathbb{E}[L] \cdot \mathbb{E}[|\xi|]) \cdot \log |\mathcal{A}|.$$

Lemma B.11 formalizes the intuition that the information a policy learns from data cannot exceed the information content it is capable of expressing in its output.

Together with Lemma B.10, this finishes the proof. $\qquad\square$

### B.6 PROOF OF THEOREM 3.3

*Proof.* To formalize this, we quantify the notion of problem complexity and establish a condition for policy failure.

**Definition B.12.** *(Required reasoning depth.) For any given query $q$, we define its Required Reasoning Depth, denoted as $L^*(q)$, as the minimum number of CoT steps required to reach any success state $m \in M_q$ from the initial state $(q, \emptyset)$. This value is an intrinsic property of the query's complexity.*

Definition B.12 can be likened to the minimum number of steps required to solve a reasoning problem. For instance, a simple two-digit addition problem might require only one or two mental steps, making its $L^*$ very low. In contrast, proving a complex theorem, such as the Pythagorean theorem, requires a sequence of many interconnected logical steps, resulting in a high $L^*$. A policy that attempts to solve the theorem with only a few steps, regardless of how smart it is, will inevitably fail.

Building on the concept of $L^*$, we now formalize the direct consequence of a policy's generated reasoning being too shallow. We make a straightforward assumption about the relationship between insufficient reasoning depth and policy failure, which translates a structural property of the CoT into a quantifiable error.

**Assumption B.13.** *(Policy failure on insufficient depth) When a policy $\pi$ generates a CoT of length $L_i$ for a query $q_i$, if the length is insufficient to solve the problem, i.e., $L_i < L^*(q_i)$, the policy is guaranteed to fail. The resulting reasoning loss $C(s_{q_i}^{\pi})$ is therefore bounded below by a significant positive constant, which we denote as $C_{fail} > 0$. In the worst case, this loss can be as high as $C_{max}$.*

Assumption B.13 claims that a policy's failure due to insufficient depth can be observed in practice. Consider a complex physics problem requiring multiple steps: identifying the forces, setting up equations, and solving for variables. A policy that only completes the first step and then stops will not be able to reach the correct final answer. In this case, the loss function would assign a high penalty (our $C_{fail}$) because the generated CoT, while possibly correct for the initial step, is fundamentally incomplete for solving the problem. The policy, therefore, cannot even fit the training data for problems that require deeper reasoning.

Building upon Definition B.12 and Assumption B.13, we can then derive a lower bound of the empirical reasoning loss $\hat{R}_n(\pi)$.

We begin with its definition:

$$\hat{R}_n(\pi) = \frac{1}{n} \sum_{i=1}^{n} C(s_{q_i}^\pi). \tag{37}$$

We can partition the sum over the training set into two disjoint subsets based on whether the generated CoT length was sufficient:

- The set of failed queries: $S_{fail} = \{q_i \in S \mid L_i < L^*(q_i)\}$.

- The set of potentially successful queries: $S_{succ} = \{q_i \in S \mid L_i \geq L^*(q_i)\}$.

Let $n_{fail} = |S_{fail}|$. We can now rewrite the empirical loss as:

$$\hat{R}_n(\pi) = \frac{1}{n} \left( \sum_{q_i \in S_{fail}} C(s_{q_i}^\pi) + \sum_{q_i \in S_{succ}} C(s_{q_i}^\pi) \right). \tag{38}$$

Applying our failure assumption, we know that for every $q_i \in S_{fail}$, $C(s_{q_i}^\pi) \geq C_{fail}$. For queries in $S_{succ}$, the loss is at least 0. This allows us to establish a lower bound:

$$\hat{R}_n(\pi) \geq \frac{1}{n} \left( \sum_{q_i \in S_{fail}} C_{fail} + \sum_{q_i \in S_{succ}} 0 \right) = \frac{n_{fail} \cdot C_{fail}}{n}. \tag{39}$$

We can define the empirical failure rate of the policy $\pi$ on the training set as $P_\pi(L < L^*) = n_{fail}/n$. This leads to our final lower bound for the empirical loss:

$$\hat{R}_n(\pi) \geq P_\pi(L < L^*) \cdot C_{fail}. \tag{40}$$

This finishes the proof. $\qquad\square$

## C  EXPERIMENTAL SETTINGS

**Dataset** The experiments are conducted on the MATH (Hendrycks et al., 2021) and GSM8K (Cobbe et al., 2021) benchmarks. The MATH training set is manually subdivided into five difficulty levels (Level 1–5) according to the original "level" field, where Level 5 is substantially more challenging than Level 1.

**Base Model** A diverse collection of pre-trained LLMs with varying architectures, parameter counts, and versions serves as the reference model for the reinforcement-learning algorithms. Specifically, the models include

- **Qwen2.5 series** (Qwen, 2025a): Qwen2.5-7B, Qwen2.5-3B, Qwen2.5-1.5B, and Qwen2.5-0.6B.

- **Qwen3 series** (Qwen, 2025b): Qwen3-8B-Base, Qwen3-4B-Base, Qwen3-1.7B-Base, and Qwen3-0.6B-Base.

- **Llama3 series** (Meta, 2024): Llama-3.1-8B-Instruct, Llama-3.2-3B-Instruct, and Llama-3.2-1B-Instruct.

The maximum response length is set to 2,048 tokens for the Qwen2.5 and Llama3 models and to 8,196 tokens for the reasoning-intensive Qwen3 models.

**Algorithm and Hyperparameters** The evaluation covers the current mainstream reinforcement-learning algorithms for mathematical reasoning: GRPO (Shao et al., 2024), PPO (Schulman et al., 2017), DAPO (Yu et al., 2025), Reinforce++ (Hu et al., 2025), and RLOO (Ahmadian et al., 2024). The reward function is rule-based (DeepSeek-AI, 2025). For PPO, the critic model is initialized from Qwen2.5-0.6B. The learning rate for critic training is $1 \times 10^{-5}$, the k3 KL-regularization coefficient is 0.001, and the GAE parameters $\lambda$ and $\gamma$ are both set to 1. For DAPO, no KL regularization is applied; the high and low clip ratios are 0.28 and 0.20, respectively, and $L_{cache}$ is $1,024$. Across all algorithms, each training sample is sampled eight times ($K = 8$). The actor is optimized with AdamW (Loshchilov & Hutter, 2017) at a learning rate of 1e-6, the training batch size is 128, and training proceeds in a purely on-policy manner for up to 50 steps. All experiments are run on $8 \times 96$ GB NVIDIA H20 GPUs.

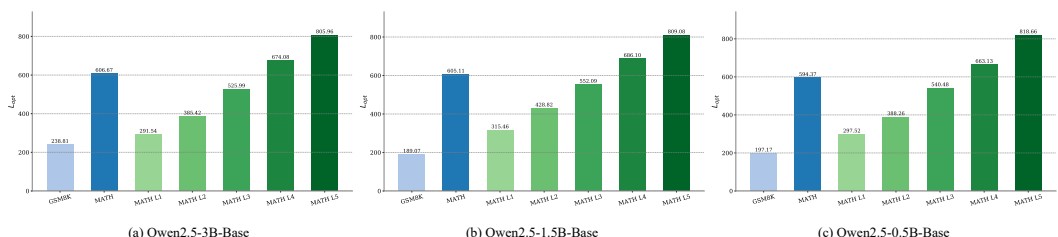

(a) Qwen2.5-3B-Base      (b) Qwen2.5-1.5B-Base      (c) Qwen2.5-0.5B-Base

Figure 6: **Supplementary validation of the relationship between task difficulty and optimal CoT length ($L_{opt}$) on additional models.** The plots illustrate that the converged mean response length ($L_{opt}$) consistently increases with task difficulty across three different models: (a) Qwen2.5-3B-Base, (b) Qwen2.5-1.5B-Base, and (c) Qwen2.5-0.5B-Base. These results reinforce the findings presented in Figure 5(a) and demonstrate the robustness of Remark 1 across models of varying capacities.

## D    MORE EXPERIMENTS

To further strengthen the empirical validation of Remark 1, we conducted supplementary experiments to verify the relationship between task difficulty and the optimal CoT length ($L_{opt}$) across models of varying capacities. The experimental protocol is identical to that described in Section 4, but here we replicate the analysis on three additional models from the Qwen2.5 series: Qwen2.5-3B-Base, Qwen2.5-1.5B-Base, and Qwen2.5-0.5B-Base.

The results, presented in Figure 6, consistently corroborate the findings from our main experiment (Figure 5(a)). Across all three smaller models, we observe the same clear, monotonic trend: the converged optimal CoT length ($L_{opt}$) increases systematically as the difficulty of the task escalates from GSM8K to the higher levels of the MATH dataset. This demonstrates that the principle outlined in Remark 1, that more complex problems necessitate a longer reasoning process to avoid underfitting is not specific to a single high-capacity model but holds true as a general principle across models of different scales.

## E    BROADER EXPLANATORY POWER OF COT-SPACE

This section further explores the extensibility of the CoT-Space framework, demonstrating its capacity to provide a unified theoretical account for several other critical and complex phenomena observed in LLMs, beyond the "overthinking" problem analyzed in the main body of this paper.

### E.1    HALLUCINATION AS TRAPPED OPTIMIZATION IN PHANTOM MINIMA

The phenomenon of hallucination (Ji et al., 2023), where a model generates fluent but factually incorrect statements, can be elegantly re-contextualized within our framework. In CoT-Space, reasoning is modeled as an optimization process on a continuous semantic manifold, aimed at minimizing a reasoning loss function $C(s)$. An ideal reasoning trajectory converges to a global minimum $M_q$, which corresponds to the correct solution. Hallucination can thus be explained as the optimization trajectory becoming trapped in a "phantom minimum" or a deep local minimum within the loss landscape. These states, while possessing a low reasoning loss due to their local coherence and syntactical correctness, do not correspond to the ground truth. Once a model's reasoning state $s_t$ enters the basin of attraction for such a trap, subsequent gradient-like updates will cause it to converge there, rather than to the true global minimum, yielding a chain-of-thought that is internally consistent but externally false.

### E.2    PROMPT SENSITIVITY AS A CONSEQUENCE OF LANDSCAPE INSTABILITY

Furthermore, the acute sensitivity of LLMs to minor, semantically irrelevant perturbations in the input prompt (Zhuo et al., 2024; Razavi et al., 2025) can be understood as an initial state perturbation problem on an unstable loss landscape. The input query $q$ defines the initial state of the reasoning process, $s_0 = (q, \emptyset)$, which serves as the starting point for the optimization trajectory. The high-

dimensional, non-convex loss landscape is likely replete with unstable topological features such as sharp ridges, crevasses, or watersheds. If an initial state $s_0$ is located near such an unstable region, a slight perturbation of the prompt (from $q$ to $q'$) can shift the initial state to $s_0'$. Although the distance between $s_0$ and $s_0'$ in the semantic space may be negligible, they might lie on opposite sides of a watershed. Consequently, their subsequent optimization trajectories will follow entirely different descent paths, converging to distinct minima and producing dramatically different outputs. This provides a formal, geometric interpretation for the model's apparent brittleness.

### E.3 Emergent Abilities as a Topological Phase Transition

The framework also offers a compelling explanation for the "emergent abilities" of LLMs, where complex reasoning capabilities appear to manifest suddenly once model scale surpasses a certain threshold, a phenomenon often described as a "reasoning cliff" for smaller models (Wei et al., 2022a). This can be interpreted as a change in the topological properties of the reasoning space itself, which is directly influenced by model capacity. Smaller models, with their limited representational power, exhibit a lower reasoning state density $\rho(K)$, as defined in Definition 2.8. As established in Theorem 2.10, a low density implies a larger expected distance between adjacent states, rendering the corresponding semantic manifold $\tilde{S}$ sparse, perforated, or even disconnected. For a complex problem, the path from the initial state $s_0$ to the solution $M_q$ may contain unbridgeable gaps, precluding any successful optimization trajectory and thus creating the reasoning cliff. As model scale increases, the representational capacity grows, leading to an exponential increase in the state density $\rho(K)$. This makes the manifold progressively more dense and smooth, effectively "filling in the holes" and establishing continuous pathways for reasoning. The emergence of an ability corresponds to a topological phase transition, where the manifold's connectivity reaches a critical point that permits a viable optimization process.

### E.4 The Efficacy of External Slow-Thinking

Finally, CoT-Space provides a principled basis for the superior performance of external slow-thinking strategies like ToT Yao et al. (2023) or MCTS Wan et al. (2024) compared to standard CoT. The difference in their efficacy can be framed as the difference between two optimization algorithms operating on the same loss landscape. Standard CoT generation is analogous to a greedy, single-path gradient descent. At each step, it commits to a single, locally optimal thought, following a solitary trajectory that is highly susceptible to being trapped in the first local minimum it encounters. In contrast, external slow-thinking strategies maintain multiple parallel candidate reasoning paths. This is equivalent to performing a beam search within the semantic state space. By exploring multiple descent directions from a given state $s_t$, external slow-thinking strategies conduct a more comprehensive exploration of the complex, non-convex loss landscape. This strategy significantly enhances the probability of circumventing local traps and identifying a path to the global optimum, thereby explaining its enhanced effectiveness on complex reasoning tasks.

## F Proposed Empirical Validation For Assumption 2.9

While Assumption 2.9 is theoretically justified by combinatorics, we propose the following method for future empirical verification. One could sample a set of abstract reasoning steps $\{\xi_i\}$ and, for each step, generate a large set of distinct token-level paraphrases using a strong LLM. By encoding these paraphrases into a semantic embedding space, one can calculate the local state density and nearest-neighbor distances. If our continuum hypothesis holds, the distribution of these distances should follow the scaling laws predicted by high-dimensional Poisson point processes as the token budget increases.

## G Prescriptive Implications

Beyond explaining current phenomena, CoT-Space provides a theoretical justification for Latent Space Reasoning. Since we prove that the reasoning space converges to a continuous manifold $\tilde{S}$, it is theoretically sound to map discrete CoT steps into this continuous latent space (e.g., via variational auto-encoders). Optimization (reasoning) could then be performed via gradient-based planning in

this smooth latent space, avoiding the discrete, non-differentiable nature of token generation, before decoding the final result. Furthermore, our framework suggests that current sparse reward models are inefficient proxies for the smooth reasoning loss $C(s)$. Future work should focus on training Continuous Verifiers that predict a scalar "semantic distance" to the solution, providing the dense, smooth gradients required for more stable RL optimization.

