# OpenReview forum: "CoT-Space: A Theoretical Framework for Internal Slow-Thinking via Reinforcement Learning"
_ICLR.cc/2026/Conference — Submitted to ICLR 2026_

### Official Review · Reviewer_so6J · 2025-10-28

**Soundness:** 2
**Presentation:** 2
**Contribution:** 1
**Rating:** 2
**Confidence:** 4

**Summary:**

This paper introduces CoT-Space, a theoretical framework for analyzing the reasoning Chain-of-Thought (CoT) of LLMs trained with RL. The authors argue that a "fundamental misalignment" exists between traditional token-level RL frameworks and the reasoning-level, multi-step nature of CoT generation, in that:
* treating the token as the action fails to capture the higher-order planning inherent to CoT, and
*  analyzing this discrete, exponentially large space with the continuous math of classical learning theory (like gradients and loss landscapes) is difficult.

The authors then present their own theory and some RL experiments that are used to support their theory, which is that
* Shorter CoTs risk underfitting by failing to reach the problem's required reasoning depth, and
* Longer CoTs risk overfitting by increasing the generalization error.

**Strengths:**

1. The experiment observations can be used to support the proposed theory.

**Weaknesses:**

1. The paper's entire motivation rests on a fundamental misalignment between traditional MDP RL and LLM CoT with RL and I don't think it is convincing. The authors use an analogy of Bricks Breaker , but this comparison is weak. The difference between an LLM's action space (a large vocabulary) and Bricks Breaker's action space ("left," "right") is one of scale and dimensionality (degrees of freedom), not a fundamental categorical difference. I can also argue that some RL environments have continous action spaces, which, in the authors' logic is also not aligned with Bricks Breaker.

2. The paper insists on a sharp divide between "token-level" actions and "reasoning-level" actions (i.e., "complete thoughts") . This distinction is arbitrary and can be applied to almost any RL problem. A Bricks Breaker agent's policy could also be broken down hierarchically (e.g., a "plan" to "hit the ball behind the bricks" , which is then executed via low-level motor commands). The paper fails to argue why CoT is fundamentally different from existing, well-studied hierarchical RL problems.

3. Combining 1 and 2, the paper claims a "conceptual gap" exists between discrete language and the "continuous mathematics underpinning classical learning theory". This ignores the vast body of RL literature that already deals with continuous action spaces. The paper attempts to re-invent the wheel and fails to situate its contribution within a standard area of RL research.

**Questions:**

See weaknesses above.

---

> ### Author Response · Authors · 2025-11-18
> **rebuttal (1/2)**
>
> We sincerely thank the reviewer for their insightful comments and critiques. The reviewer's comments highlight a core thesis in our paper that requires significant clarification. We believe the reviewer's skepticism about our premise (the "fundamental misalignment") stems from a core misunderstanding: The reviewer has misinterpreted our argument as being about the scale of the action space, whereas our core contribution is the mathematical justification for the topology of the state space and its continuum approximation.
>
> We will address each of the reviewer's perceived weaknesses point by point:
>
> ### Regarding Weakness 1: The Bricks Breaker Analogy & Continuous Action Spaces
> We appreciate the critique and want to clarify our simple intent. The Bricks Breaker analogy (Fig 1) was solely intended to illustrate one concept: that **RL, guided only by a high-level reward, can enable a policy to autonomously discover an effective, emergent strategy**. The analogy was simply: Just as a game agent can discover a good tactic (like "hitting the ball behind the bricks") on its own, an LLM policy can discover a good strategy (like "applying a suitable CoT") on its own. The reviewer is correct that the state spaces and complexities are vastly different. However, these differences were not the point of the analogy. The reviewer's focus on this discrepancy misses the simple, high-level point we were making about strategy discovery.
>
> This confusion, however, leads us directly to the core problem our paper does address, which is not about analogies but about the fundamental nature of the state space:
>
> 1. **The key issue is not the action space, but the nature of the state space**: In both Bricks Breaker (discrete actions) and a task like MuJoCo (continuous actions), the state space (e.g., ball position, brick status, joint angles) is typically low-dimensional and possesses a smooth physical/geometric structure. The transition from $s_t$ to $s_{t+1}$ is governed by simple physical laws.
> 2. **An LLM's Token-Level State Space is Fundamentally Different**: In a traditional LLM MDP, the state $s_t$ is a token sequence. The state $s_{t+1}$ is simply $s_t$ plus one new token. At a semantic level, $s_t$ and $s_{t+1}$ are nearly equivalent. It takes a vast number of token-level actions to complete a single meaningful "reasoning step."
> 3. **Our Contribution (CoT-Space)**: Our paper does not claim the LLM's action space is continuous. On the contrary, we point out that the token-level state space is discrete, high-dimensional, and topologically complex (Assertion 2.1). Applying classical learning theory (e.g., analyzing loss landscapes) directly in this space is intractable. Therefore, our core contribution is to mathematically justify and construct a continuous "semantic state space" as an approximation. The existing literature on "continuous action space RL" is orthogonal and unrelated to our contribution, which is a continuum approximation of the LLM state space.

---

> ### Author Response · Authors · 2025-11-18
> **rebuttal (2/2)**
>
> ### Regarding Weakness 2: "Token-Level vs Reasoning-Level" Distinction & HRL
> We thank the reviewer for connecting our framework to Hierarchical Reinforcement Learning (HRL). This is a valuable perspective. However, we argue there is a critical, non-arbitrary mathematical distinction between CoT-Space and standard HRL, which is the core of our paper (Section 2.3).
> 1. Limitations of Standard HRL: In the reviewer's Bricks Breaker HRL example, a high-level "plan" (e.g., "hit the ball behind the bricks") still maps to one (or a few) specific low-level action sequences. The high-level "state" is still the physical state of the game.
> 2. The Fundamental Difference of CoT-Space (Exponential Expressive Redundancy of Language): Our "Reasoning-Level" abstraction is not arbitrary; it is grounded in the semantic equivalence of language (Definition 2.7).
>     - As shown in Figure 3(a), a single "Reasoning-Level" step (our "high-level plan"), e.g., "calculate the total weekly pages," can correspond to an exponential number of semantically equivalent, yet token-level-distinct, sequences (e.g., $\xi_{i_1}, \xi_{i_2}, \xi_{i_3}$...).
>     - This is the cornerstone of our paper: Assumption 2.9 (Exponential expressive redundancy of language).
> 3. Why This Isn't Standard HRL: It is precisely this "many-to-one" mapping (from token sequences to semantic states) that makes the underlying state space extremely dense when "zoomed out."
>     - Theorem 2.10 (Continuum convergence) rigorously proves that this density allows the "Reasoning-Level state space" to be approximated as a continuous manifold.
>     - This continuum approximation, grounded in semantic redundancy, is a property that standard HRL (in Bricks Breaker or robotics) does not possess. We are not just proposing a hierarchy; we are providing a mathematical proof of continuity for the semantic state space of that hierarchy.
>
> To our best knowledge, CoT is fundamentally different from HRL framework. We kindly request the reviewer to list some specific HRL works which can be used to explain LLM reasoning for further discussion.
>
> ### Regarding Weakness 3: Ignoring Continuous RL & "Re-inventing the Wheel"
> With all due respect, this is a complete misreading of our contribution.
>
> 1. We are not "ignoring" continuous mathematics; we are "building a bridge" to it. The root problem is this: LLM reasoning is inherently a discrete, token-by-token generation process (per Assertion 2.1). Classical learning theory (involving gradients, continuous loss landscapes, etc.) cannot be directly applied to this discrete space.
>
> 2. CoT-Space is the Bridge: The entire point of our paper (in Section 2.3) is to be the first to mathematically and rigorously justify why this discrete, token-level process can be approximated as an optimization process in a continuous semantic space (CoT-Space).
>
> 3. How we achieve this: It is precisely through "semantic equivalence" (Def 2.7) and "exponential expressive redundancy" (Assum 2.9) that we prove the "continuum convergence" (Thm 2.10) of the reasoning state space.
>
> 4. Conclusion: We are far from "re-inventing the wheel." **We are building a "connecting axle"** that links the "discrete language generation" wheel to the "classical continuous optimization theory" wheel.
>     - It is this bridge (CoT-Space) that allows us, in Section 3, to analyze CoT length ($L_{opt}$) using the tools of continuous mathematics, like optimization noise (Thm 3.1) and generalization risk (Thm 3.2, 3.3), in a principled way for the first time.
>     - These analyses are impossible within a purely discrete, token-level MDP framework.
>
> We strongly believe CoT-Space provides a missing and critical theoretical foundation for understanding LLM reasoning in RL. We respectfully urge the reviewer to reconsider our core contribution: the mathematical justification for the continuum approximation of the LLM reasoning state space, and the bridge it builds between discrete reasoning and continuous learning theory. Hence, we hope the reviewer can reconsider our paper's contribution.

---

### Official Review · Reviewer_thdN · 2025-11-01

**Soundness:** 1
**Presentation:** 1
**Contribution:** 1
**Rating:** 2
**Confidence:** 4

**Summary:**

The paper proposes CoT-Space, a reasoning-level framework for RL-trained LLMs that reframes CoT generation as optimization over a continuous semantic manifold. Section 2 defines reasoning states, posits exponential “expressive redundancy” (Assump. 2.9), and proves density-driven continuum results. Section 3 analyzes “overthinking” via noise and generalization-risk, predicting an optimal CoT length Lopt. Section 4 empirically explores correlations between Lopt and task difficulty, model capacity, RL algorithm, and KL loss.

**Strengths:**

As of now, I tried but cannot sense any strengths, though the intention to provide a scaffold to achieve reasoning-level formalism is clear

**Weaknesses:**

- authors use $L$ for both token-length and step-depth. In Sec. 3.1 $L$ denotes CoT length; in Sec. 3.2 $L$ denotes the number of reasoning steps; Fig. 5 treats $L_{opt}$ as length, but the related text for fig 5 links the trend to Theorem 3.3 in terms of reasoning depth $L^*$. This notational conflation makes Theorem 3.1, the empirical proxies in Fig. 5, and the bounds in 3.2 difficult to interpret consistently.

- Reasoning-level is treated as a “simple extension” from tokens to steps, but the paper assumes the output is segmented into steps $\xi_1$,…,$\xi_L$ (Defs. 2.2–2.6; Secs. 2.2–2.3) without specifying a segmentation rule or identifiability guarantees. This is a core barrier in practice: semantic step decomposition is non-unique and task/model dependent. Section 2 builds on this premise, and Section 3’s analysis and bounds depend on it, but no operational definition, annotator protocol, or robustness analysis is provided.

- Key constructs are under-specified: Decode(·) and semantic equivalence “≡” in Def. 2.7, the “semantic volume” Vsemantic and its metric in Def. 2.8, and the dimensionality D and manifold assumptions in Theorem. 2.10–2.12. Without a metric and regularity conditions (e.g., smoothness/Lipschitz continuity of C, curvature bounds), the continuum arguments are under-anchored.

- Noise–length relation (Theorem 3.1) lacks a precise definition of g in the main text. If g is an effective optimisation-noise scale analogous to SGD noise, how is it operationalized for CoT generation? Further, where is the relation used in the paper? any purpose and why? if KL divergence is the proxy, justify theoretically why KL tracks the relevant noise scale.

- Theorem 3.2 adapts supervised-learning results to RL-style reasoning but omits RL-specific caveats: policy-induced distribution shift, dependence between samples due to on-policy rollouts, and reward-driven selection effects; we may need to clarify the learning setting (on/off-policy, i.i.d. queries vs trajectories)

- Lower bound (Theorem 3.3) formalizes underfitting via $P_{\pi}$ (L < L*), but L* (minimum required depth) is not observable. How is  L* estimated, and how sensitive are conclusions to estimation error?

- Fig. 5’s empirical “validations” largely echo patterns already reported elsewhere (task difficulty ↔ longer CoT; stronger models ↔ shorter CoT). What is new here relative to prior empirical studies on CoT length vs difficulty/capacity?


- Several broad claims need tighter support in main text. For example: “token-level state space is problematic” (Sec. 2.1), and “zooming out allows abstract, stepwise analysis” (Sec. 2.2 figure narrative) are asserted without formal counterexamples or empirical diagnostics; “density arises from the expressive redundancy of language” (Sec. 2.3) is not plausible
- Redundancy: Sec. 2.1 vs Sec. 2.2 and Fig. 2 repeat the token vs reasoning-level distinction with similar prose. This space could instead be used for a concrete worked example grounding Def. 2.2–2.6.
- The link of this work to “internal slow-thinking via RL” is underdeveloped in the main. Beyond the analogy (Fig. 4a–c), the paper does not define slow thinking rigorously nor show how a specific RL objective operationalizes it.

- if the novelty is the proposed theoretical justification; however, due to the issues above (notation, assumptions, theorem gaps), the empirical section reads as confirmation of known trends rather than a stress-test of your theory.

**Questions:**

- What does “classical theoretical results” refer to precisely in line 305 and Sec. 2.3 end? Which specific theorems (e.g.,   information-theoretic bounds with mutual information) are you invoking, and under which assumptions do they carry over?
- What is a “suitable CoT” in Fig. 1(b)?
- You aim to understand internal slow-thinking via RL, but where do you define slow thinking rigorously?
- What precisely is “noise” in Theorem 3.1 and Fig. 5(d)? If KL loss is the proxy, why is it a valid stand-in for optimization noise in reasoning-space?
- How do you estimate L* in practice for Theorem 3.3, and how robust are results to misestimation?
- For Def. 2.7 (semantic equivalence set), how do you operationalize Decode(·) and “≡” beyond intuition? Is it exact logical equivalence, or model-judged equivalence with tolerance?

---

> ### Author Response · Authors · 2025-11-18
> **rebuttal (1/4)**
>
> We sincerely thank you for your detailed review and the insightful feedback on our paper. Your sharp comments have highlighted several core issues that require clarification and improvement.
>
> We believe many of the concerns you raised, particularly regarding the definition of "reasoning steps," the rigor of the "semantic space," and the link between theory and practice, stem from a potential misunderstanding of our work's central objective. **Our goal is not to provide a complete, immediately computable or empirically measurable model. Rather, our primary aim is to build a "conceptual bridge",** as stated in the abstract, to **connect classical machine learning (ML) theories (e.g., optimization, generalization risk) to the reasoning phenomena observed in LLMs**, which currently lack a strong theoretical foundation.
>
> We aim to "revitalize"  these classical theories, arguing for their continued relevance in the LLM era. To do this, we posit several core concepts (like reasoning steps and semantic equivalence) and, from them, derive a self-consistent theoretical framework. The value of this framework lies in its ability to ***explain*** key empirical phenomena, such as "overthinking".
>
> Based on this, we will provide a detailed response to each of your Weaknesses and Questions.
>
> ## Response to Weaknesses
>
> ### W1: Notational Conflation of $L$
> We wholeheartedly agree. We are thankful for your valuable suggestions which will help us improve the clarity of our manuscript.
> In the revised manuscript, we will enforce a strict separation:
> 1. $L$: Will uniquely refer to the "number of reasoning steps" (Step Depth). This aligns with the context of Thm 3.1 (noise-length relation) and Thm 3.3 (underfitting risk).
> 2. $K$: We will introduce (and globally unify) $K$ to represent the "total token count" (Token Length) (as already used in Appendix B.1, e.g., $K=kL$).
> 3. Thm 3.2 (Generalization Bound): This bound is indeed related to the total token count $K$ (specifically, $\mathbb{E}[L] \cdot \mathbb{E}[|\xi|]$), as this reflects the model's "information capacity" or "complexity."
> 4. Fig. 5: The $L_{opt}$ in the charts empirically measures "Response Length," which we use as a proxy for both $K$ and $L$. Also, our theoretical explanation for this trend relies on the trade-off involving both $L$ (Thm 3.3 for underfitting) and $K$ (Thm 3.2 for overfitting).
>
> By clearly distinguishing $L$ (steps) from $K$ (tokens), our theory will precisely articulate the trade-off between "underfitting" (driven by $L$) and "overfitting" (driven by $K$), making the entire framework much clearer.
>
> ### W2: Segmentation of Reasoning Steps
> This is a core theoretical assumption of our paper, and we appreciate the chance to clarify it. We do not view this as a "simple extension," but as the fundamental abstraction of our framework.
>
> 1. Our goal is a **Reasoning-Level** theory. Just as classical physics models an object as a "point mass" (at the macro-level) without  considering its atomic composition, our framework posits the existence of conceptual steps $(\xi_1, \xi_2, ...)$. The very idea of CoT is premised on this intuition of "steps."
> 2. Our theory **does not depend on a unique, computable segmentation rule**. It only requires that some semantically coherent decomposition exists. The theory's value is in showing if we analyze at this abstract level (the intuitive level of CoT), what conclusions can we draw.
> 3. Our contribution is to show that, by accepting this abstraction, we can model reasoning as an optimization trajectory in a continuous semantic space, thereby applying classical learning tools. How to operationally find these steps is a challenging empirical question, but it is outside the scope of our ***theoretical*** paper.
>
> ### W3: Under-specified Constructs
> This is closely related to W2. These constructs are **formal concepts** used to build the theoretical framework.
> 1. Decode($\cdot$) and “$\equiv$” (Def 2.7): Their purpose is not to provide a computable function, but to formally define the concept of a "semantic equivalence set" $\mathcal{V}(\xi_l, k)$. This is the bedrock for our core assumption, "Exponential expressive redundancy" (Assump. 2.9).
> 2. V_semantic and $D$ (Def 2.8): Similarly, we do not need to specify the metric or exact dimension $D$ of this space. Our key argument (Thm 2.10) is that regardless of the specific metric or $D$, as long as Assump. 2.9 holds, the state density $\rho(K)$ grows exponentially ($\rho(K) = \Theta(c^K)$).
> 3. The exponential growth of $\rho(K)$ forces the expected nearest neighbor distance $\mathbb{E}[dist(s_i, s_j)] \propto \rho(K)^{-1/D}$ to rapidly approach zero. This proves the convergence to a continuum. It is this convergence, not a prior assumption of smoothness, that licenses our (conceptual) use of continuous optimization tools (like a reasoning loss $C$ and its gradient).

---

> ### Author Response · Authors · 2025-11-18
> **rebuttal (2/4)**
>
> ### W4: Noise–length relation (Thm 3.1)
> 1. Definition of $g$: We apologize for the lack of clarity in the main text. $g$ is the Noise Scale in the Stochastic Differential Equation (SDE) model of the reasoning process, which we introduce in Appendix B.4.
> 2. The theorem ($g \propto 1/L$) is derived theoretically by equating the variance from the discrete gradient update (Eq. 29) and the continuous SDE model (Eq. 30), as shown in Eq. 36. It establishes a fundamental inverse relationship between reasoning depth $L$ and optimization noise $g$.
> 3. Justification of KL Proxy (Fig 5d): In RL, the KL divergence (often from a reference policy) is a standard measure of the "aggressiveness" or "variance" of a policy update. We use the KL-Loss (the policy's KL to its own previous version) as an empirical proxy. We hypothesize that an update process with a high KL-Loss is "noisier" and more "aggressive," corresponding empirically to the higher noise scale $g$ in our theoretical model.
>
> ### W5: RL-specific caveats (Thm 3.2)
> 1. Scope and Assumptions: Thm 3.2 (which relies on Lemma B.10) is indeed a classical information-theoretic generalization bound adapted from Statistical Learning Theory (SLT). We explicitly assume i.i.d. sampled queries.
> 2. Purpose: Our goal here is to analyze the generalization risk of the final, converged policy as a function of its output complexity (i.e., token length $K$). The SLT bound is the most direct tool for this.
> 3. We acknowledge that a full analysis of the training dynamics of an RL algorithm (like on-policy PPO) would need to account for distribution shift. However, **our analysis is focused on the overfitting risk of the resulting policy** (its performance on an i.i.d. test set), which is driven by its complexity. Thm 3.2 captures this. We view this as a valuable first-order approximation of the overfitting risk. We will clarify this assumption and scope in the revision.
>
> ### W6: Unobservability of $L^*$
> Just like our response in W2 and W3, $L^*$ is indeed unobservable, but its purpose is not estimation.
> 1. $L^*$ (Def B.12) is a **theoretical construct**, defined as the intrinsic minimum number of steps required to solve a problem.
> 2. The purpose of Thm 3.3 is not to compute an exact bound, but to theoretically prove the existence of an "underfitting risk". It formalizes the intuition that "if a task requires $L^*$ steps, any policy producing $L < L^*$ steps is guaranteed to fail."
> 3. This theorem (underfitting), combined with Thm 3.2 (overfitting), rigorously proves that the total error curve must have a U-shaped (or similar) trade-off, which in turn proves the existence of an optimal $L_{opt}$. Our experiment (Fig 5a) is not designed to measure it, but to validate the prediction derived from its existence: that as task complexity increases (presumed $L^*$ increases), the observed $L_{opt}$ will also increase.
>
> ### W7: Empirical Validations are Known Trends
> We strongly disagree with this characterization.
> 1. Observation vs. Explanation: Prior work observed these trends, but they remained largely isolated empirical phenomena.
> 2. Our framework is the first to provide a unified theoretical explanation for all of these trends. Our theory models them as a fundamental trade-off between underfitting (Thm 3.3, explaining ***why*** $L_{opt}$ must increase with difficulty) and overfitting (Thm 3.2, explaining ***why*** $L_{opt}$ decreases with model capacity to manage generalization risk).
> 3. Our experiments are not for discovery; they are for validation. The fact that our theory successfully predicts these previously unexplained trends is strong evidence for our theory's validity.
>
> ### W8: Broad Claims Need Tighter Support
> We appreciate the call to strengthen these arguments.
> 1. "Token-level is problematic" (Sec 2.1): This is supported by Assertion 2.1 (exponential growth and discrete topology ). It is precisely this discreteness and exponential growth that hinders the direct application of classical (continuous) analytical tools.
> 2. "Zoom out" (Sec 2.2): This is a narrative description of the consequence of our core result, Thm 2.10 (continuum convergence). Because we proved the space becomes continuous (macroscopically), we can (theoretically) perform a stepwise (i.e., gradient-like) analysis.
> 3. "Density from redundancy" (Sec 2.3): This is the central hypothesis of our paper, explicitly stated in Assumption 2.9 (Exponential expressive redundancy). This assumption that one semantic step can be realized by exponentially many token sequences, is the "axiom" on which our theory is built. We believe it is highly plausible for natural language. We will revise the text to make it clearer that this is our foundational assumption.

---

> ### Author Response · Authors · 2025-11-18
> **rebuttal (3/4)**
>
> ### W9: Redundancy
> This is a very good suggestion. Our intent was to use Sec 2.1 to pose the problem (the token-level dilemma) and Sec 2.2 to propose the solution (the reasoning-level CoT-Space ). We agree the prose can be streamlined. We will merge the redundant descriptions and use the saved space, as you suggested, to add a concrete example illustrating Defs. 2.2–2.6.
>
> ### W10: Link to "internal slow-thinking via RL"
> This is related to Q3.
> 1. Definition: "Internal slow-thinking" is a background of our research. We define "internal slow-thinking" (line 39, 43) as the model internally generating a multi-step CoT, distinguishing it from external methods like tree-search. This branch of studides are also known as "test-time scaling".
> 2. Role of RL: "via RL" refers to this internal process being optimized via a high-level reward signal.
> 3. Role of CoT-Space: Our framework, CoT-Space, models the state space in which this internal process operates. Figure 4(b) explicitly models the "Cycle of LLM Reasoning" as a "State" optimization driven by a "Policy" (the RL agent).
>
> We will strengthen this link in the introduction.
>
> ### W11: Novelty
> We believe our responses to W1-W10 demonstrate that the perceived "flaws" are primarily misunderstandings of our paper's theoretical scope. Our novelty lies in:
> 1. Proposing CoT-Space as a novel abstract framework.
> 2. First proof of the reasoning space's convergence to a continuum (Thm 2.10).
> 3. First unified explanation of $L_{opt}$ trends (difficulty, capacity) via the classical underfitting/overfitting trade-off (Thm 3.2, 3.3).
>
> By clarifying the notation (W1) and the (non-computational) nature of our theoretical constructs (W2, W3, W6), we are confident the value and novelty of our theory will be clear.
>
>
> ## Response to Questions
>
> ### Q1: What does “classical theoretical results” refer to precisely...?
> We are referring primarily to two branches of classical theory.
> 1. Optimization Theory (Sec 3.1): Concepts like loss landscapes, gradient descent, optimization noise, and Sharpness-Aware Minimization (SAM).
> 2. Statistical Learning Theory (Sec 3.2): Specifically, PAC-Bayesian analysis (Thm B.8 ) and information-theoretic generalization bounds (Thm 3.2, from Lemma B.10), which link generalization error to mutual information and model complexity.
> They "carry over" because our Thm 2.10 justifies treating the reasoning space (CoT-Space) as a continuous manifold, which allows the reasoning loss $C(\cdot)$ to be analyzed analogously to a classical loss function.
>
> ### Q2: What is a “suitable CoT” in Fig. 1(b)?
> A "suitable CoT" refers to a chain-of-thought that leads to the correct final answer. In our framework, this corresponds to a trajectory that successfully converges to a "minimum" (Def 2.3) in the reasoning loss landscape. In the context of Figure 1, the RL agent discovers that generating this correct, multi-step, and CoT with suitable length is the "good strategy" to maximize its reward (getting the correct answer).
>
> ### Q3: ...where do you define slow thinking rigorously?
> (Related to W10) We define "slow-thinking" (line 39, 43) broadly as any process that elicits multi-step reasoning (like CoT), as opposed to "fast-thinking" (immediate, single-step answers). We further categorize it into:
> * External: e.g., tree-search, self-consistency.
> * Internal: e.g., the model learns to generate the CoT itself via post-training.
>
> ### Q4: What precisely is “noise” in Theorem 3.1 and Fig. 5(d)?
> (Related to W4)
> * In Thm 3.1: "Noise" refers to the theoretical noise scale $g$ in our SDE model of the optimization process (Appendix B.4).
> * In Fig. 5(d): The "proxy" is the KL-Loss. We use this as an empirical proxy for $g$, based on the rationale that a more "aggressive" or high-variance update (high KL) in RL corresponds to a "noisier" optimization process.

---

> ### Author Response · Authors · 2025-11-18
> **rebuttal (4/4)**
>
> ### Q5: How do you estimate $L$ in practice for Theorem 3.3...?
> (Related to W6) We do not estimate $L^*$.
> * $L^*$ is a theoretical construct (Def B.12) representing the intrinsic minimum steps a problem requires.
> * Its purpose in Thm 3.3 is solely to prove the existence of an "underfitting risk", which in turn proves the necessity of the $L_{opt}$ trade-off.
>
> ### Q6: For Def. 2.7... how do you operationalize Decode($\cdot$) and “$\equiv$”...?
> We do not operationalize them.
> * They are formal constructs. Their purpose is to define the concept of the "semantic equivalence set" $\mathcal{V}(\xi_l, k)$.
> * The only requirement for our theory is that this concept leads to Assumption 2.9 (Exponential expressive redundancy)—i.e., that one semantic step $\xi$ can be expressed by exponentially many token sequences k.
> * This assumption, which we find highly plausible for language, is all that is needed to derive our key continuum result (Thm 2.10). Our theory does not require a computable Decode function.
>
>
> ## Conclusion
> We thank the reviewer again for the insightful (though critical) comments. We acknowledge the oversights in notational rigor (W1) and clarity (W9), which we will fix decisively in the revision.
>
> However, on the more fundamental criticisms (W2, W3, W6, W7), we believe there is a core misunderstanding of a **theoretical paper** (which seeks to explain) versus an empirical paper (which seeks to measure). Our contribution is a conceptual framework that, through reasonable assumptions, brings the nascent phenomenon of LLM reasoning under the umbrella of classical learning theory, providing the first unified theoretical explanation for the $L_{opt}$ trends. We are confident that with these clarifications, the value and novelty of our work are evident.
>
> We respectfully ask the reviewer to reconsider our contribution in light of these clarifications.

---

### Official Review · Reviewer_ogAP · 2025-11-01

**Soundness:** 3
**Presentation:** 3
**Contribution:** 3
**Rating:** 6
**Confidence:** 3

**Summary:**

This manuscript proposes a theoretical framework, CoT-Space, that optimizes chain-of-thought (CoT) reasoning in a continuous semantic space rather than at the discrete token level. It formalizes reasoning states and minima, proves a continuum approximation under an expressive redundancy assumption, and analyzes overthinking from both optimization-noise and information-theoretic perspectives. Experiments on math tasks indicate that the optimal CoT length Lopt increases with task difficulty and decreases with model size.

**Strengths:**

- Clear motivation and conceptual framing that highlight the misalignment between token-level RL and reasoning-level CoT.

- Detailed definitions and derivations (with supporting appendices), which improve readability and rigor.

**Weaknesses:**

- Assumption 2.9 (exponential semantic redundancy) is central to the continuum result but is not empirically validated, especially given the structured nature of LLM-induced semantic spaces.

- Definition 2.6 provides only an abstract ordering of the loss consistent with reachable distances and does not offer a concrete instantiation.

- Empirical details are insufficiently presented.

**Questions:**

- In the experiments, what concretely proxies C(⋅) (e.g., task-level reward, verifier loss)? How sensitive are your conclusions to this choice? How to choose a proper C in general?

- Beyond theory, can you approximate “state density” by sampling semantically equivalent steps and measuring nearest-neighbor distances in a task-conditioned embedding, to empirically test the continuum assumptions and derived conclusions?

- It seems that the takeaways from the framework deduction are limited (to a suitable CoT length regarding task difficulty), can this framework extend to support a direct CoT optimization in a latent space, e.g., assisting in finding this continuous space and construction loss?

---

> ### Author Response · Authors · 2025-11-18
> **rebuttal (1/2)**
>
> We sincerely thank you for your thoughtful and constructive review of our paper. Your insightful critiques regarding Assumption 2.9, Definition 2.6, and the framework's future potential are invaluable, and we believe addressing them will significantly strengthen our paper. We are pleased that you found our conceptual framing to be a key strength. We will now address each of your specific weaknesses and questions in turn.
>
> ### Weakness 1 & Question 2: On Assumption 2.9 (Exponential Redundancy) and Empirical Validation
> We thank the reviewer for this insightful critique.
> - Theoretical Justification: Our primary goal was to establish a theoretical framework. Assumption 2.9 serves as a formal condition under which the continuum approximation holds. We posit that this assumption is reasonable due to the combinatorial nature of language: a single abstract reasoning step (e.g., "calculating the weekly pages") can be expressed by a vast and exponentially growing number of token-level realizations (as illustrated in Figure 3a). While the high-level semantic space (e.g., the "concept" of addition) may be structured by pre-training, the expression of reasoning steps within that structure still allows for enormous local redundancy.
>
> - Empirical Validation (Q2): We strongly agree with your excellent suggestion for an empirical test. Your proposal to approximate "state density" by sampling semantically equivalent steps (e.g., via paraphrasing with a strong LLM) and then measuring nearest-neighbor distances in an embedding space is a highly promising and methodologically sound direction for future work.
>
> This is a non-trivial empirical investigation in its own right, which is why we focused on the theoretical implications in this paper. However, the fact that our framework inspires such concrete, testable hypotheses is one of its key strengths.
>
> We will revise the discussion of Assumption 2.9 in Section 2.3 to be more precise. We will explicitly state that it is a foundational posit for our theoretical results. Furthermore, we will add a new paragraph to the appendix (or the main discussion, if space permits) to address this point directly, outlining the exact empirical validation strategy you suggested as a clear and valuable path for future research.
>
> ### Weakness 2 & Question 1: On Definition 2.6 (Abstract Loss $C(\cdot)$) and its Proxy
> This is a critical point of clarification.
>
> 1. Abstract by Design: You are correct that Definition 2.6 is abstract. The purpose of this definition is not to construct a specific loss function, but to establish the properties that any such function must have in a reasoning space (i.e., it must decrease as we approach a valid solution). Our theoretical framework is built to be general, holding true for any loss function that satisfies this basic ordering.
>
> 2. Experimental Proxy (Q1): In our experiments, the abstract loss $C(\cdot)$ is not directly observed. Instead, we observe a sparse, high-level signal: the task-level reward (i.e., a binary reward for a correct final answer).
>
> The RL algorithm's value function (in PPO, for example) is precisely the component that learns to approximate the expected future reward from any given state. This learned value function is, in effect, the algorithm's empirical, learned approximation of our theoretical reasoning loss $C(\cdot)$. The RL agent, by maximizing its reward, is implicitly performing optimization (descending the gradient) on this unobserved loss landscape.
>
> 3. Sensitivity and General Choice (Q1): This is an excellent question. Our framework contextualizes this choice.
>     - Choice: A sparse, task-level reward (as we used) is the simplest proxy. A more sophisticated proxy, as you suggest, could be a fine-grained verifier loss.
>     - Sensitivity: Our core theoretical takeaways (e.g., the underfitting/overfitting trade-off that leads to $L_{opt}$) are a consequence of the optimization dynamics themselves, not the specific proxy. As long as the proxy (task reward, verifier loss) correctly identifies the "minimum" (the correct answer), the fundamental tension between taking enough steps to reach the minimum (avoiding underfitting, Thm 3.3) and not taking so many steps that you overfit (avoiding high generalization error, Thm 3.2) will still hold. We hypothesize the specific value of $L_{opt}$ might change, but not its existence.
>
> We will add a paragraph to Section 2.2 to explicitly bridge this gap between theory and practice. We will clarify that $C(\cdot)$ is a theoretical construct and that in our RL experiments, the (implicit) value function learned from the (explicit) task-level reward serves as its practical approximation.

---

> ### Author Response · Authors · 2025-11-18
> **rebuttal (2/2)**
>
> ### Weakness 3: Insufficient Empirical Details
> We apologize if the details were difficult to locate. We placed the full experimental setup (models, datasets, algorithms, hyperparameters) in Appendix C to maintain the flow of the main paper. To address your concern, we will move a more detailed summary of the key setup (e.g., the specific models and task levels used for Figure 5) from the appendix into the main body of Section 4.
>
> ### Question 3: On Limited Takeaways and Extending the Framework
> This is a profound question that hits on the future-looking value of our work.
>
> 1. Descriptive vs. Prescriptive Power: Our paper's primary contribution is descriptive: to build the first, to our knowledge, theoretical framework that moves beyond the token-level and successfully explains observed empirical phenomena (like "overthinking" and the existence of $L_{opt}$) by unifying them with classical learning theory (underfitting/overfitting, optimization noise). We believe this descriptive power is a significant contribution in itself.
>
> 2. Prescriptive Potential (The Extension): You are exactly right that the framework's ultimate promise is prescriptive. Our framework is not just an explanation; it is a foundation. By modeling reasoning as an optimization problem in a continuous space, it directly motivates and provides a theoretical basis for the very extensions you propose:
>     - Direct Latent Space Optimization: Our work provides the theoretical justification for why one would want to do this. A future system could, for example, use an auto-encoder to learn a mapping from discrete CoT steps to our continuous CoT-Space, and then perform RL or planning directly in that latent space, which would be far more efficient and smoother than optimizing in the discrete token space.
>     - Constructing Better Losses: Our framework suggests that the "ideal" reasoning loss $C(\cdot)$ should be smooth. This guides the design of better reward models or verifiers: instead of just outputting a binary score, they should be trained to produce a continuous "distance-to-solution" estimate, providing a much denser and more effective training signal.
>
> This is the most exciting avenue for future work. We will expand our Conclusion and Appendix E ("Broader Explanatory Power") to explicitly discuss this prescriptive potential, including the two directions you've helped us articulate: 1) learning a continuous CoT-Space for direct latent optimization and 2) using the framework to guide the design of non-sparse, continuous loss functions for reasoning.
>
> We thank you again for your time and expertise. Your feedback has been instrumental in helping us see the path to a much-improved paper. We believe that by incorporating these clarifications, especially regarding the role of Assumption 2.9, the proxy for $C(\cdot)$, and the prescriptive future of the framework, we can fully resolve your concerns and elevate the paper's contribution.

---

### Official Review · Reviewer_pW19 · 2025-11-02

**Soundness:** 3
**Presentation:** 3
**Contribution:** 3
**Rating:** 6
**Confidence:** 2

**Summary:**

This paper proposes CoT‑Space, a reasoning‑level view of RL for LLMs that replaces token‑level MDPs with states, reachable minima, and a reasoning loss, thereby recasting multi‑step reasoning as optimization in a semantic space rather than discrete token control. It further argues this space becomes approximately continuous as the token budget grows and bounds the discrete–continuous gap, which licenses gradient/SDE analyses. Building on this, the paper gives two lenses for the empirically observed optimal CoT length: a noise view and a risk view. Experiments on GSM8K/MATH and multiple model families/algorithms support four predictions: increases with task difficulty, decreases with model capacity, is largely algorithm‑agnostic, and inversely correlates with a KL‑loss proxy for noise.

**Strengths:**

The paper offers a clear conceptual bridge from token‑level RL to reasoning‑level optimization with concrete definitions and a continuum justification, enabling learning‑theory tools to address CoT phenomena. The two complementary analyses provide a unified explanation for both under‑thinking and over‑thinking and yield testable predictions that are validated across tasks, model scales, and RL methods.

**Weaknesses:**

1. Idealized assumptions. The existence of a semantics-aware loss and reachable minima for queries with a single "golden" answer narrows scope and is non-constructive; the continuum argument relies on unmeasured semantic density/dimension, making it hard to falsify.

2. Simplified noise model. The SDE derivation treats noise as white and state-independent to obtain $g \propto 1 / L$; mapping KL-loss to "noise" is only a proxy and may conflate effects.

3. Loose generalization bounds. PAC-Bayes/information-theoretic bounds are qualitative (constants and mutual-information terms not estimated) and assume i.i.d. data, whereas training is on-policy.

**Questions:**

See weakness

---

> ### Author Response · Authors · 2025-11-18
> **rebuttal (1/2)**
>
> We thank the reviewer for the detailed feedback and positive assessment of our paper's core contributions, particularly in offering a "clear conceptual bridge" and a "unified explanation for both under-thinking and over-thinking." We are encouraged that the reviewer found our predictions to be "validated across tasks, model scales, and RL methods."
>
> We appreciate the insightful questions, we hope to resolve these ambiguities and demonstrate the soundness of our theoretical approach.
>
> ### W1: On Idealized Assumptions
> We thank the reviewer for this crucial point. We acknowledge that our framework relies on abstractions, which we argue are necessary for building a tractable theoretical model of the complex reasoning process.
> - On Non-Constructive Loss and Minima: This is a deliberate theoretical choice, analogous to the use of "loss landscapes" in classical deep learning analysis. While the exact, high-dimensional loss function of a neural network is "non-constructive," its conceptual existence (Definition 2.6) allows us to apply powerful analytical tools (e.g., from optimization theory). Our assumption is mild: we only assume that some reasoning states are "closer" to a correct solution than others. The "golden answer" scope (Definition 2.3) is a standard and necessary simplification for analyzing reasoning in tasks like GSM8K and MATH, which are the primary benchmarks in this field.
> - On the Continuum Argument: The reviewer correctly notes that measuring the "semantic dimension" is difficult. However, the value of our continuum approximation (Section 2.3) is not in its direct measurement, but in its predictive power. Our framework posits that if the reasoning space becomes continuous (driven by the "exponential expressive redundancy of language" in Assumption 2.9), then a fundamental trade-off between underfitting and overfitting must emerge, leading to an optimal CoT length ($L_{opt}$).
>
> The "falsifiability" of our theory thus lies not in measuring the dimension, but in testing its predictions. Our extensive experiments in Section 4 do exactly this: we test the four key predictions derived from our theory (regarding $L_{opt}$ vs. task difficulty, model capacity, algorithm, and noise) and find they all hold true. This provides strong empirical validation for the utility of our abstract model.
>
> We will clarify in the final version that these definitions are intended as conceptual tools for theoretical analysis, not as objects to be directly computed.
>
> ### W2: On the Simplified Noise Model
> This is an excellent point regarding the trade-off between model simplicity and precision.
> - On the SDE Simplification: The simplification to white, state-independent noise (Section 3.1 and Appendix B.4) is a standard technique in theoretical modeling, allowing us to derive a clear, first-order relationship between the CoT length ($L$) and the effective noise scale ($g$). The goal of Theorem 3.1 is to reveal the inverse correlation ($g \propto 1/L$), which provides the theoretical basis for the "overthinking" phenomenon from a noise perspective. We believe this core insight would hold even with a more complex noise model, which would likely obscure this fundamental relationship.
> - On KL-Loss as a Proxy: We agree with the reviewer that KL-Loss is a proxy for the optimization noise ($g$). However, we argue it is a well-motivated one. In RL algorithms like PPO, the KL term explicitly controls the policy's deviation from a reference policy, which acts as a form of optimization regularization. This deviation is a primary source of "noise" in the optimization trajectory. Our empirical finding in Figure 5(d), which shows a clear inverse correlation between KL-Loss and the converged $L_{opt}$, validates that this proxy, while not perfect, is empirically meaningful and strongly supports our theoretical claim from Theorem 3.1.
>
> We will add a discussion to Section 4 to make the motivation for this proxy, and its limitations, more explicit.

---

> ### Author Response · Authors · 2025-11-18
> **rebuttal (2/2)**
>
> ### W3: On the Generalization Bounds
> We sincerely thank the reviewer for this sharp and insightful technical point. We have used these established bounds as the most direct theoretical tools available to conceptualize and model the risk of overfitting. Our primary objective in Section 3.2.1 was not to derive a new, technically-tight bound for on-policy learning, which is a highly complex and open research area in itself. Instead, our goal was to build a principled argument for why "overthinking" (i.e., excessively long CoTs) can be mechanistically understood as a form of overfitting. The terms identified by these bounds (e.g., CoT length $L$ and verbosity $|\xi|$ in Theorem 3.2) serve as a powerful conceptual proxy for policy complexity, allowing us to connect the structural properties of the reasoning process to generalization risk for the first time.
>
> We position our work as a preliminary but, we believe, pioneering exploration into a new theoretical area. The main contribution of CoT-Space is to provide the "conceptual bridge"  that allows these classical theories to be applied to LLM reasoning at all. We will explicitly add this discussion to the limitations in our final manuscript and highlight the development of more rigorous, non-stationary bounds as a critical direction for future work building on our framework.

---

### Meta-Review · Area_Chair_uFX4 · 2026-01-08

**Summary:**

The paper introduces CoT-Space, a theoretical framework that models Chain-of-Thought (CoT) reasoning in LLMs not as a discrete token-level Markov Decision Process (MDP), but as an optimization process over a continuous, reasoning-level semantic manifold. The authors posit that due to the "exponential expressive redundancy" of language (many token sequences map to one semantic thought), the state space becomes dense enough to approximate a continuum. Using this framework, the authors apply classical learning theory (optimization landscapes, noise analysis, generalization risk) to explain empirical phenomena, such as the trade-off between underfitting (short CoT) and overfitting (long CoT), and the existence of an optimal CoT length correlated with task difficulty and model capacity.

The major concern raised by the reviewers was about the theoretical rigor of the paper. For example, as pointed out by ogAP, the core premise of the paper, "Exponential Expressive Redundancy" which justifies the continuum approximation, lacked empirical validation. The definition of "semantic equivalence" and "reasoning steps" remained abstract without operational definitions (e.g., a segmentation rule). Reviewers ogAP and pW19 questioned how theoretical constructs like "reasoning loss" and "noise scale" map to practical metrics. While the authors use proxies (Task Reward, KL Divergence), the link between the abstract theory and these proxies was not strong enough.

**Reviewer Concerns:**

While the authors have addressed some concerns, such as the conflation of the notations, some core concerns remained unaddressed, especially the reliance on unverified assumptions (redundancy) and the loose application of i.i.d. generalization bounds to RL weakened the formal contributions.

**Reviewer Scores:**

The reviewers' scores were unlikely to change much, given that the core concerns remained.

---

### Decision · Program_Chairs · 2026-01-26

Reject